# Electrocaloric effects across room temperature in multilayer capacitors

M. Guo[1 ✉], V. Farenkov[1], X. Chen[1], A. Mohanathan[2], A. Z. K. Goh[2], Y. Tang[1], J. Zhang[1], M. Vickers[1], S. M. Fairclough[1], C. Ducati[1], X. Moya[1 ✉], S. Hirose[3 ✉] & N. D. Mathur[1 ✉]

A growing number of cooling devices[1–4] exploit large electrocaloric effects associated with a supercritically driven first-order ferroelectric phase transition in multilayer capacitors of $PbSc_{0.5}Ta_{0.5}O_3$ (PST)[5]. However, these multilayer capacitors only operate above the room-temperature Curie temperature and require an energetically expensive 42-day anneal for high B-site order to maximize latent heat. Here we show that exaggerating valence mismatch through dilution with $PbMg_{0.5}W_{0.5}O_3$ (PMW) maintains high B-site order and latent heat with no anneal, while disrupting dipolar order to reduce the Curie temperature as low as 230 K. Our multilayer capacitors of PST–PMW show supercritical electrocaloric effects of about 3 K across and well below room temperature owing to 17.1 V μm$^{-1}$ fields we apply >$10^7$ times without breakdown. Using our multilayer capacitors in an ideal fluid regenerator and assuming work recovery yields cycle efficiencies of 70–90%. Taken together, our findings imply that multilayer capacitors of PST–PMW should now replace multilayer capacitors of PST in electrocaloric prototypes to permit electrocaloric refrigeration.

Highly reversible thermal changes can arise when ferroelectric phase transitions are driven by applied electric fields above the Curie temperature $T_C$. These electrocaloric (EC) effects[6,7] are typically observed near and above room temperature because most ferroelectrics have relatively high Curie temperatures[8,9]. Moreover, EC effects can be large in thin layers as breakdown fields can be large[10–14] and assemblies of thin ceramic/polymer layers (multilayer capacitors (MLCs))[1–4,15–24], and flexible polymer bilayers[25–29]/monolayers[30,31], have been exploited in a growing number of EC prototypes that cool loads down to temperatures near room temperature. However, EC cooling of a load through room temperature remains elusive, even though such cooling is important for food, beverages, the built environment, medicine and so on.

EC effects in existing material systems cannot be exploited for cooling through room temperature. For PST modified by B-site disorder[32] or solid solution[33], sub-room-temperature EC effects were reported in thin single layers and are small (≤1.6 K). For other ceramics, sub-room-temperature EC effects are sometimes larger (≤1–4 K) but they arise in thin single layers[14] or decay to zero and change sign on crossing room temperature—as seen for antiferroelectric PMW[34–36]—such that it is thermodynamically impossible to use PMW to cool to or through room temperature (Supplementary Note 1) (ref. 14 also demonstrated an MLC that was compromised by substantial inactive thermal mass and only measured at room temperature).

Here we describe large-active-volume unannealed MLCs based on until now unexplored solid solutions of $(1 − x)$PST–$x$PMW, in which PMW disruption of PST dipolar order suppresses $T_C$ without overly disrupting B-site order, such that the latent heat of the ferroelectric phase transition remains large[37]. Supercritically driving this transition yields large EC effects over a wide range of operating temperatures that extends above and well below room temperature.

Preliminary screening in $0.05 \leq x \leq 0.25$ led us to select $x = 0.15$ (85PST–15PMW) and $x = 0.10$ (90PST–10PMW) to maintain EC performance while suppressing $T_C$ as measured on heating down to 230 K and 242 K, respectively. The transition is strongly first order in our solid solutions because B-site order[37] between high-valence cations ($Ta^{5+}$, $W^{6+}$) and low-valence cations ($Sc^{3+}$, $Mg^{2+}$)—observed using X-ray diffraction (XRD) and high-angle annular dark-field scanning transmission electron microscopy (HAADF-STEM)—is facilitated by valence/size cation mismatch and achieved after low-temperature sintering without the long, expensive anneal. Despite inactive thermal mass in the active area (outermost PST–PMW layers, inner electrodes), the effective temperature change is large ($|\Delta T_{eff}| \approx 3$ K), and similar to repeatably driven temperature changes in the MLC working bodies[5] of EC prototypes[1–4].

Our measurements of EC temperature change, performed with a small thermocouple, are consistent with indirect measurements based on dense adiabatic electrical polarization data and heat capacity data[38]. These indirect measurements are used to construct entropy maps on which we identify entropy changes consistent with direct measurements of isothermal EC heat (the high-field leakage is so low that even our sensitive calorimetry does not detect Joule heating except at very high temperatures).

If MLCs of PST[5] in prototypes[1–4] were replaced with MLCs of PST–PMW and likewise driven with 600 V, one could cool down to near 230 K, not 295 K, and slightly increase efficiency as seen by permuting variables in our entropy maps and constructing balanced Brayton-like cycles in which one or more MLCs traverses an ideal fluid regenerator[38].

[1]Department of Materials Science, University of Cambridge, Cambridge, UK. [2]Department of Physics, University of Cambridge, Cambridge, UK. [3]Murata Manufacturing Co., Ltd., Kyoto, Japan. ✉e-mail: mg2129@cam.ac.uk; xm212@cam.ac.uk; h_sakyo@murata.com; ndm12@cam.ac.uk

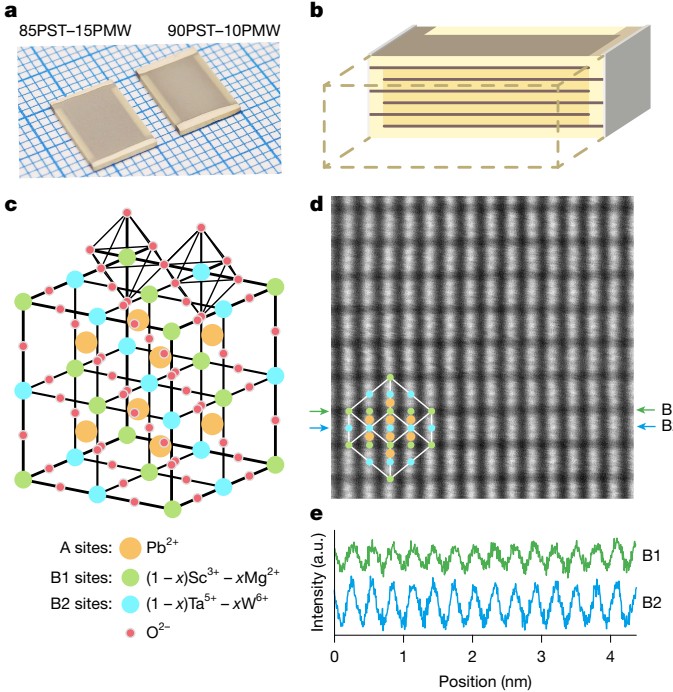

**Fig. 1 | Structure of MLCs and PST–PMW. a**, Optical image showing MLCs with 19 active layers. **b**, Not-to-scale simplified schematic showing half an MLC that comprises active PST–PMW (yellow), inactive PST–PMW (pale yellow), interdigitated Pt inner electrodes (brown) and Ag terminals (grey). **c**, Unit cell of fully ordered PST–PMW in the cubic (c) phase. **d**, HAADF-STEM image of 85PST–15PMW viewed down one of the $<211>_c$ directions. Overlay identifies unit cell (white outline), A-site cations ($Pb^{2+}$), B1-site cations with low $Z$ ($Sc^{3+}$, $Mg^{2+}$) and B2-site cations with high $Z$ ($Ta^{5+}$, $W^{6+}$). **e**, Line profiles for the arrowed rows in **d** show B-site columns every 0.30 nm. Data in **d** and **e** for MLC1 of 85PST–15PMW. a.u., arbitrary units.

We suggest that these improvements could trigger an explosion of industrial and academic R&D into EC cooling.

## MLCs

As described in Methods, we fabricated 19-layer MLCs of 85PST–15PMW and 90PST–10PMW (Fig. 1a). MLC geometry is presented qualitatively in Fig. 1b and quantitatively in Methods. Figure 1c shows the cubic (c) crystal structure with fully ordered B-site cations (B1 and B2) that form a chequerboard in the $\{200\}_c$ planes. XRD reveals that both compositions have similar lattice parameters and good B-site order between high-valence and low-valence cations: 8.13 Å and $S_{111} \approx 0.99$ for 85PST–15PMW and 8.14 Å and $S_{111} \approx 0.97$ for 90PST–10PMW (Supplementary Note 2), where $S_{111} = ((I_{111}/I_{200})_{measured}/(I_{111}/I_{200})_{expected})^{1/2}$ compares measured and expected intensity $I$ values for the 111 and 200 reflections[5,37].

Good B-site order between high-valence and low-valence cations in 85PST–15PMW is confirmed by HAADF-STEM. A representative image (Fig. 1d) shows a region with the zone axis along one of the $<211>_c$ directions. The B1 columns (for example, between green arrows) and the B2 columns (for example, between blue arrows) are atomically resolved and line profiles along these arrowed columns (Fig. 1e) show well-resolved peaks whose 0.30-nm separation matches well the value of 0.29 nm obtained from the lattice parameter. Supplementary Note 3 explores other zone axes.

The intensity of a given atomic column in a HAADF-STEM image depends on the average value of $Z^{1.7}$ assuming constant sample thickness[39] ($Z$ is atomic number). In Fig. 1d, the B2 columns are on average 136% more intense than the B1 columns, confirming that B2 columns

comprise high-$Z$ elements ($Ta^{5+}$ with $Z = 73$, $W^{6+}$ with $Z = 74$) and that B1 columns comprise low-$Z$ elements ($Sc^{3+}$ with $Z = 21$, $Mg^{2+}$ with $Z = 12$), as expected given the good B-site order between high-valence and low-valence cations identified with XRD. Although the B1 columns contain $Sc^{3+}$ and $Mg^{2+}$ ions with rather different atomic numbers, similar intensities suggest good mixing.

The thermally hysteretic transition observed in heat capacity data reveals that increasing PMW content, and thus disorder, reduces absolute transition temperatures and increases thermal hysteresis (Supplementary Note 4). Dielectric spectra reveal that this thermal hysteresis coexists with a relaxor behaviour owing to PMW disruption of PST (Supplementary Note 5).

## EC measurements

EC effects associated with the active PST–PMW are initially evaluated using the indirect (Maxwell) method based on adiabatic polarization-field measurements[38] $P(E)$, as set out below for 85PST–15PMW and in Supplementary Note 6 for 90PST–10PMW. The $P(E)$ plots were obtained using rapid bipolar sweeps out to $E = \pm 17.1 \text{ V } \mu m^{-1}$ (600 V) after heating to 322 values of set temperature $T_s$ that are separated by 0.45 K and lie mainly above $T_C \approx 230 \text{ K}$ (determined from zero-field calorimetry data measured on heating; Supplementary Note 4). Positive field-removal branches show good reversibility (Supplementary Note 7) and are used to construct a map of $P(T_s, E)$ and thus $P(S', E)$ (Fig. 2a), in which $T_s$ (nonlinear bottom axis) is converted to the relative entropy $S'(T_s) = S(T_s) - S(200 \text{ K})$ for zero-field heating (Supplementary Note 4) ($S$ denotes absolute entropy).

Isofield differentiation of $P(S', E)$ was performed after data smoothing (Supplementary Note 8) and before data truncation (all Fig. 2 panels omit data for the 21 lowest and 17 highest set temperatures). The resulting map of $-(\partial P/\partial S)_E$ (Fig. 2b) is combined with the Maxwell relation[38] $(\partial T/\partial E)_S = -(\partial P/\partial S)_E$ to obtain the nominally reversible adiabatic temperature change $\Delta T(S', E) = -\int_0^E (\partial P/\partial S)_{E'} dE' \geq 0$ for field application (Fig. 2c).

Adding $\Delta T(S', E)$ to the zero-field temperature $T(S', 0)$, which matches the set temperature and is therefore identified by inverting $S'(T_s)$ and letting $T = T_s$, yields a map of absolute temperature $T(S', E)$ (Fig. 2d). Permuting variables in $T(S', E)$ yields a map of relative entropy $S'(T, E)$ (Fig. 2e) from which we identify the nominally reversible isothermal entropy change $\Delta S(T, E) = S'(T, E) - S'(T, 0)$ for field application (Fig. 2f). This computationally easy method for obtaining $\Delta S(T, E)$ (Fig. 2f) is conceptually equivalent to following the field-driven displacement of isothermal contours along the entropy axis in $T(S', E)$ (Fig. 2d). The phase boundary (dotted pink line in all Fig. 2 panels; see Methods) ends at a room-temperature critical point (299 K, 5.0 V $\mu m^{-1}$), and above and below this temperature large EC effects can be driven using supercritical fields.

Extended Data Table 1 reports indirectly measured peak EC effects of $|\Delta T| \approx 4 \text{ K}$, $|Q| = T|\Delta S| \approx 9 \text{ MJ m}^{-3}$ and $|\Delta S| \approx 34 \text{ kJ K}^{-1} \text{ m}^{-3}$ for both PST–PMW compositions with 600 V ($Q$ denotes isothermal EC heat). The EC entropy change here exceeds the entropy change from the electrically driven transition alone, which is nominally equivalent to the thermally driven zero-field entropy change for the transition ($|\Delta S_0| \approx 14 \text{ kJ K}^{-1} \text{ m}^{-3}$ for both PST–PMW compositions; Supplementary Note 4). Given that $|\Delta S|$ exceeds $|\Delta S_0|$ by 160% (85PST–15PMW) and 120% (90PST–10PMW), whereas the corresponding figure is 23% for PST ($|\Delta S| \approx 43 \text{ kJ K}^{-1} \text{ m}^{-3}$, $|\Delta S_0| \approx 35 \text{ kJ K}^{-1} \text{ m}^{-3}$)[5], EC effects in the untransformed and transformed phases are relatively pronounced in PST–PMW (Supplementary Note 9).

Direct measurements of isothermal EC heat are described below for 85PST–15PMW and in Supplementary Note 10 for 90PST–10PMW. These measurements were performed using a bespoke calorimeter[40,41] that permits electrical access to our large MLCs (Methods). At 24 temperatures set sequentially on heating, we measured the

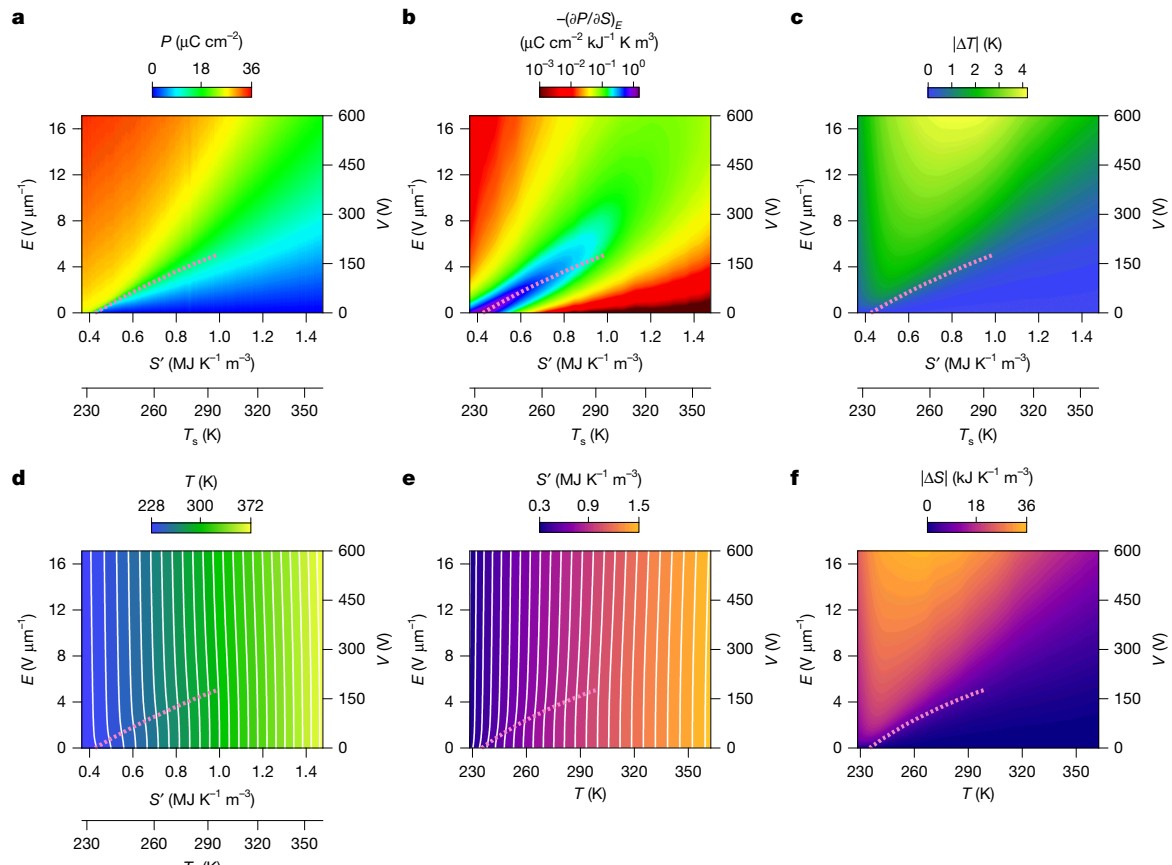

**Fig. 2 | Indirect EC measurements of 85PST–15PMW. a,** Polarization $P(S', E)$ comprising highly reversible field-removal branches ($E \geq 0$) of bipolar $P(E)$ plots measured adiabatically out to $E = \pm 17.1$ V µm⁻¹ after heating to each set temperature that we identify at zero field as $T_s$. Each value of $T_s$ is converted to relative entropy $S'$ at zero field by means of the heating branch of $S'(T) = S(T) - S(200 \text{ K})$ with $T = T_s$ (Supplementary Note 4). **b,c,** Using $P(S', E)$ data obtained over a slightly wider range of set temperatures, we obtain $-(\partial P/\partial S)_E$ (**b**),

which, through the indirect method, yields $|\Delta T(S', E)|$ (**c**). **d,** Adding $T_s(S')$ at each value of $S'$ yields absolute temperature $T(S', E)$, isothermal contours every 5.8 K. **e,** Permuting variables yields relative entropy $S'(T, E)$, adiabatic contours every 48 kJ K⁻¹ m⁻³. **f,** Subtracting $S'(T)$ at each value of $T$ yields $|\Delta S(T, E)|$. Smoothed phase boundary (dotted pink line) ends at critical point. Voltage $V$ corresponds to field $E$. Data for MLC1 of 85PST–15PMW.

active-volume-normalized heat per unit time $dQ(t)/dt$ during application and removal of $E = 21.4$ V µm⁻¹ (750 V) over 3,000 s (Supplementary Note 11). This field change is highly isothermal given a much smaller thermal relaxation time of 10 s (Methods). Division by $|dE(t)|/dt$ yields the differential heat $dQ(E)/|dE|$ with good fidelity at intermediate temperatures (Fig. 3a) (Supplementary Note 12 shows and explains poor-fidelity $dQ(E)/|dE|$ plots at high and low temperatures and Supplementary Note 13 compares the good-fidelity $dQ(E)/|dE|$ plots with indirect predictions). Figure 3a shows that increasing temperature upshifts the transition field along the phase boundary (Fig. 2e). The concomitant shortening and broadening of the $dQ(E)/|dE|$ peaks is associated with approaching the critical point and the aforementioned EC effects in the untransformed and transformed phases.

The magnitude of the isothermal EC heat $|Q|$ was evaluated at all measurement temperatures by integrating heat flow $dQ(t)/dt$ during the application and removal of $E = 21.4$ V µm⁻¹ (Fig. 3b). Figure 3c shows the corresponding values of $|\Delta S| = |Q|/T$. Values of $|Q|$ are similar for field application and field removal and we identify peak values of $|Q| \approx 10$ MJ m⁻³ for both PST–PMW compositions (Extended Data Table 2). Figure 3b also shows values of $|Q|$ that we obtained by integrating the same $dQ(t)/dt$ data during the times over which $E = 8.6$ and 17.1 V µm⁻¹ were applied and removed. Physically applying (removing) such fields should give identical (similar) results.

At our highest measurement temperatures, field-application values of $|Q|$ slightly exceed field-removal values owing to a small Joule heating during our slow isothermal measurements. This small Joule

heating is apparent from our sensitive calorimetry measurements when holding at $E = 21.4$ V µm⁻¹ and dissipation is the same order of magnitude as dissipation identified from steady-state leakage (Methods). Indirectly measured values of $|Q|$ (solid lines in Fig. 3b) match well with directly measured values despite relaxor behaviour (Supplementary Note 5).

Direct measurements of EC temperature change are described below for 85PST–15PMW and in Supplementary Note 10 for 90PST–10PMW. These measurements were performed using a thermocouple at the MLC face centre (Methods) (infrared imaging is challenging below room temperature). For 29 increasing values of set temperature $T_s$, we report temperature-jump magnitude $|\Delta T_j|$ on applying and subsequently removing $E = 8.6$, 17.1 and 21.4 V µm⁻¹ (left axis of Fig. 3d, example of raw data in the Fig. 3d inset, all raw data in Supplementary Note 14). Values of $|\Delta T_j|$ are similar for field application and field removal and we identify peak values of roughly $|\Delta T_j| \approx 3$ K for both PST–PMW compositions (Extended Data Table 2). At our highest (lowest) measurement temperatures, field-application values are slightly larger (smaller) than field-removal values because the isofield legs in the Brayton-like cycle are nonlinear, as explained for MLCs of PST in ref. 5.

A least-squares fit between the values of $|\Delta T_j(T_s)|$ that we measured directly when removing $E = 8.6$ V µm⁻¹ and the corresponding values of $|\Delta T(T_s)|$ (solid lines in Fig. 3d, right axis) that we identified for the active volume using the indirect method (Fig. 2c) implies $|\Delta T_j| \approx f|\Delta T|$ with $f = 0.72$. Here $f$ is the product of a factor $f_1$ for initial layer thermalization in the active area and a factor $f_2$ for subsequent thermalization of

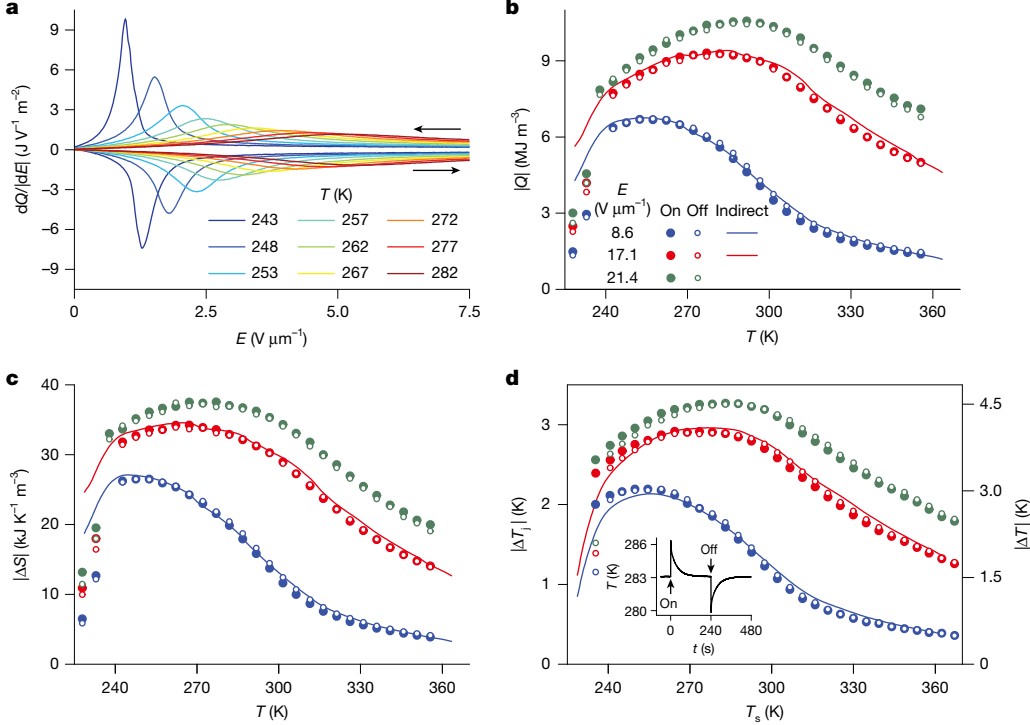

**Fig. 3 | Direct measurements of EC heat and temperature change for 85PST–15PMW. a**, Differential heat d$Q$/|d$E$| versus field $E$ identified through the isothermal application and subsequent removal of $E = 21.4$ V μm$^{-1}$ at 9 of 24 measurement temperatures, data normalized by active volume and plotted out to 7.5 V μm$^{-1}$. **b,c**, |$Q$| (**b**) and |$\Delta S$| = |$Q$|/$T$ (**c**) versus temperature $T$ for application (solid symbols) and removal (open symbols) of $E = 21.4$ V μm$^{-1}$, with data inferred for $E = 8.6$ V μm$^{-1}$ and 17.1 V μm$^{-1}$. Solid lines show the corresponding indirect data from Fig. 2f. **d**, Directly measured temperature jump (|$\Delta T_j$|, left axis) and

indirectly measured temperature change (|$\Delta T$| from Fig. 2c, right axis) versus set temperature $T_s$ for the three fields. Supplementary Note 14 shows all raw data, inset shows example of raw data for $T_s = 283$ K and $E = 21.4$ V μm$^{-1}$, $t$ denotes time. We identified |$\Delta T_j$| = 0.72|$\Delta T$| through a least-squares fit to field-removal |$\Delta T_j(T_s)$| data with $E = 8.6$ V μm$^{-1}$. All data in **a**–**d** obtained on heating. Data derived from calorimetry for MLC2 of 85PST–15PMW. All other data for MLC1 of 85PST–15PMW.

the active area with the thermocouple and the affixing drop of black paint. Given that geometry and volumetric heat capacity imply $f_1 = 0.85$ (Methods), we deduce $f_2 = f/f_1 \approx 0.85$. The |$\Delta T_j$| ≈ $f$|$\Delta T$| scaling in Fig. 3d implies that |$\Delta T$| for the active layers reaches 4.5 K.

## Effective EC temperature change

Figure 4 shows the range of the effective temperature change |$\Delta T_{eff}$| that can be achieved with MLCs of 85PST–15PMW (blue data) and 90PST–10PMW (green data) when applying and removing a voltage (600 V) for which there exists corresponding data for geometrically similar MLCs of PST (purple data[5]) (Supplementary Note 15 shows |$\Delta T_{eff}$| data for our maximum voltage of 750 V). At a given set temperature $T_s$, the upper bound (|$\Delta T_{eff}$| ≈ |$\Delta T_j$|/$f_2$) assumes useful heat transfer between the thermalized active area and, say, regenerator fluid, whereas the lower bound (|$\Delta T_{eff}$| ≈ $v_{active}$|$\Delta T$| ≈ $v_{active}$|$\Delta T_j$|/$f$) assumes thermalization of the entire MLC before useful heat exchange (active volume fraction $v_{active} = 57$–58%; Methods).

Even away from the upper thermalization limit, MLCs of both PST–PMW compositions show large and highly reversible EC effects above and well below room temperature down to near 230 K, unlike MLCs of PST, which perform well only above room temperature. The maximum values of |$\Delta T_{eff}$| are 3.4 K (85PST–15PMW), 3.2 K (90PST–10PMW) and 4.3 K (PST), implying an approximately 20% reduction from PMW dilution. A typical magnetocaloric working body (bed of commercial-grade Gd spheres), driven by a magnetic flux density of 1.4 T from permanent magnets, shows relatively small magnetocaloric effects over a relatively narrow range of temperatures[5,42] (red line in Fig. 4). Magnetocaloric working bodies based on other materials show similar/inferior performance (Supplementary Note 16).

## EC cooling cycles

Here we consider limiting upper bounds on performance by describing cooling cycles (Fig. 5a,b) in which one or more MLCs, driven using ≤600 V and assuming 100% work recovery[21] (>99% has been

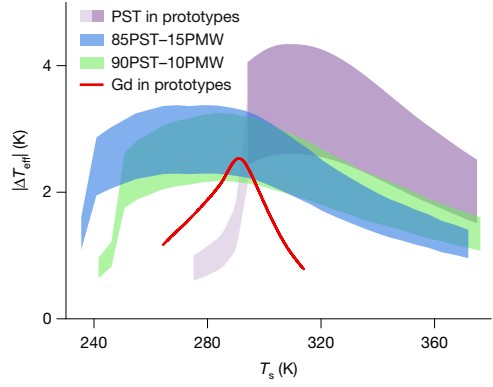

**Fig. 4 | Effective EC temperature change above and below room temperature.** Effective EC temperature change |$\Delta T_{eff}$| versus set temperature $T_s$ for geometrically similar MLCs driven by voltage changes of 600 V. Upper bounds assume thermalization of active area, lower bounds assume thermalization of entire MLC. Data based on field-off measurements of |$\Delta T_j$| from Fig. 3d (85PST–15PMW, $E = 17.1$ V μm$^{-1}$), Supplementary Fig. 15d (90PST–10PMW, $E = 16.7$ V μm$^{-1}$) and Fig. 4 of ref. 5 (PST, $E = 15.8$ V μm$^{-1}$), except pale purple data are based on indirect measurements in Fig. 2 of ref. 5 (PST, $E = 15.8$ V μm$^{-1}$). Red data[5,42] represent a bed of commercial-grade Gd spheres in which magnetocaloric effects are driven using 1.4 T.

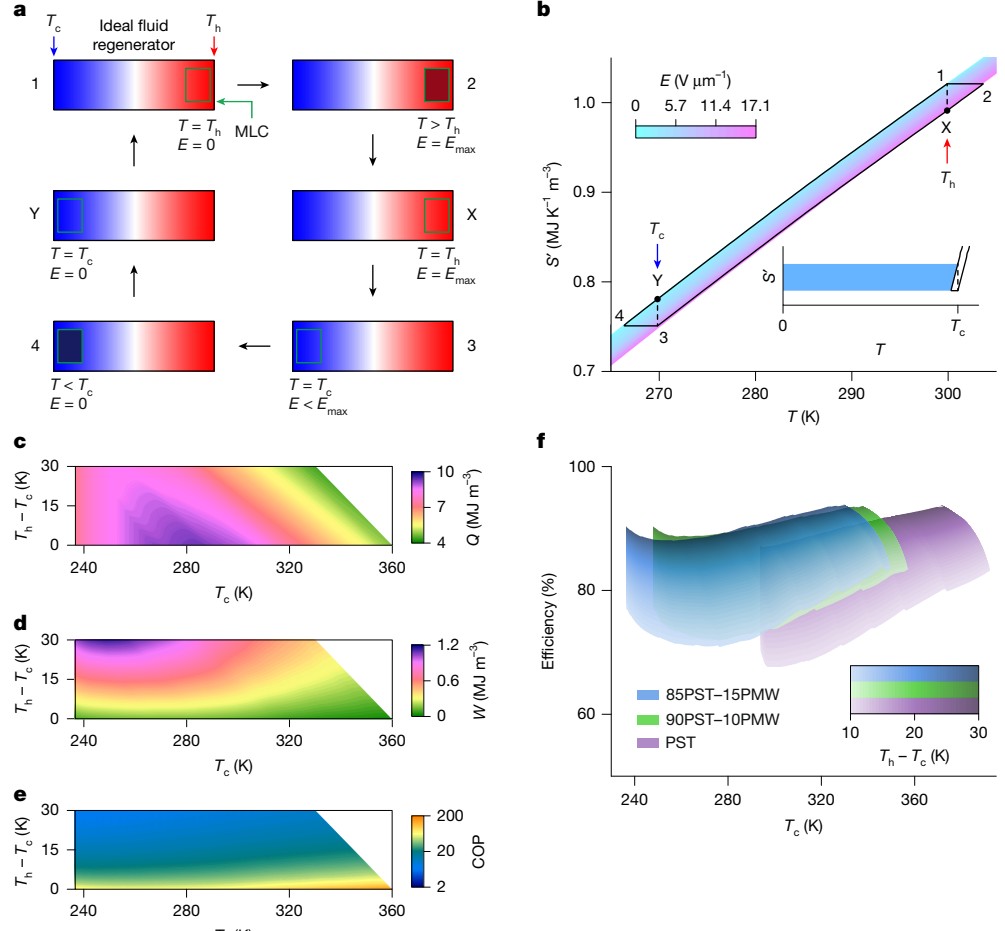

**Fig. 5 | Efficiency of regenerative cooling cycles. a,b**, Schematics show an MLC translated between hot (red, $T_h$) and cold (blue, $T_c$) ends of an ideal fluid regenerator during balanced Brayton-like cycles ($1 \to 2 \to X \to 3 \to 4 \to Y \to 1$) (**a**) of the type shown on an $E(T, S')$ map (**b**) that represents the active volume in an MLC of 85PST–15PMW. In **b**, dashed vertical lines identify $T_c = 270$ K and $T_h = 300$ K. Inset identifies the heat $Q$ (blue) absorbed from a load during $4 \to Y$. **c–f**, Using the same $E(T, S')$ map in a wider temperature range (Supplementary

Note 17), we evaluate the heat $Q$ (**c**), cycle work $W$ (**d**), COP = $Q/W$ (**e**) and cycle efficiency (**f**) as functions of cold temperature $T_c$ and temperature span $T_h - T_c$. Panel **f** also shows the corresponding data for geometrically similar MLCs of 90PST–10PMW and PST (ref. 5) driven using 600 V. All data derived from $E(T, S')$ maps obtained by permuting the variables in $S'(T, E)$ maps ($T > T_c$) that appear in Fig. 2e (85PST–15PMW, $E \leq 17.1$ V μm$^{-1}$), Supplementary Fig. 10e (90PST–10PMW, $E \leq 16.7$ V μm$^{-1}$) and Fig. 2c of ref. 5 (PST, $E \leq 15.8$ V μm$^{-1}$).

demonstrated[24]), undergo Brayton-like cycles ($1 \to 2 \to X \to 3 \to 4 \to Y \to 1$) that involve translation ($X \to 3$, $Y \to 1$) between the hot and cold ends of an ideal fluid regenerator at $T_h$ and $T_c$ ($T_c$ differs from Curie temperature $T_C$). The $E(T, S')$ map for MLCs of 85PST–15PMW ($E \leq 17.1$ V μm$^{-1}$; Fig. 5b) forms part of a larger map (Supplementary Note 17) that was obtained by permuting the variables in $S'(T, E)$ (Fig. 2e) above $T_C$ to avoid ferro-electric hysteresis (Supplementary Note 7). The $E(T, S')$ map permits the construction of Brayton-like cycles that are balanced owing to field variation during regenerator transit $X \to 3$ (ref. 38) (Supplementary Note 18). Although the $E(T, S')$ map was obtained from measurements of active MLC volume, it can be taken to represent the entire MLC, as regenerator operation assumes thermalization between active EC material and local surroundings, such that inactive MLC volume may be thermally subsumed into nearby fluid.

The active-volume-normalized heat $Q = \int_4^Y T \mathrm{d}S'$ (Fig. 5c) pumped from a load at the cold end of the regenerator (blue area, inset of Fig. 5b) varies with $T_c$ and also temperature span $T_h - T_c$ given the balanced regeneration[38]. The maximum heat ($|Q| = 9.3$ MJ m$^{-3}$) necessarily falls just short of the maximum isothermal heat ($|Q| \approx 10$ MJ m$^{-3}$; Fig. 3b) and corresponds to a maximum cooling power of 77.7 mW per MLC given our active volume (32.6 mm$^3$; Methods) and a typical operational frequency (0.25 Hz) that is limited by dielectric fluid rather than ceramic thermal conductivity[3].

Cycle area gives the corresponding net work done $W = \oint T \mathrm{d}S'$ (Fig. 5d). Dividing the resulting COP = $Q/W$ (Fig. 5e) by the Carnot limit $T_c/(T_h - T_c)$ yields cycle efficiency (Fig. 5f), which we present along with the corresponding data for our MLCs of 90PST–10PMW ($E \leq 16.7$ V μm$^{-1}$) and MLCs of PST ($E \leq 15.8$ V μm$^{-1}$)[5]. Supplementary Notes 17 and 20 show maps of $E$, $Q$, $W$, COP and efficiency for all three MLC compositions.

If MLCs of PST–PMW replace MLCs of PST (ref. 5), then prototypes[1–4] should operate with slightly improved efficiencies (Fig. 5f) owing to small concomitant reductions in EC effects (Fig. 4), as explained in Supplementary Note 19 (lowering operating voltages likewise increases efficiencies; Supplementary Notes 20 and 21). Our efficiencies range from about 70% ($T_h - T_c = 10$ K) to about 90% ($T_h - T_c = 30$ K), the latter exceeding the former mainly because increasing $T_h - T_c$ reduces the Carnot limit.

## Discussion and outlook

We have developed MLCs in which partial substitution of PST with PMW permits densification by means of low-temperature sinter-ing, thus preserving B-site cation order between high-valence and low-valence cations and obviating a long, expensive anneal[5]. The pres-ervation of B-site order limits the suppression of latent heat, whereas

the disruption of dipolar order reduces the Curie temperature above which large EC effects can be supercritically driven over a wide range of temperatures.

PST diluted with PMW has good B-site order between high-valence and low-valence cations because the valence mismatch between $Sc^{3+}$ (0.75 Å) and $Ta^{5+}$ (0.64 Å) in B-site-ordered PST is exaggerated in the replacement cations $Mg^{2+}$ (0.66 Å) and $W^{6+}$ (0.60 Å) and because the corresponding size mismatch is partially preserved. To understand why good B-site order between high-valence and low-valence cations is possible with no anneal, first consider MLCs of PST[5] and PMW[35]. MLCs of PST require a 42-day anneal to restore the pre-existing B-site order destroyed by the high-temperature sintering (1,400 °C) required for densification and thus high breakdown fields. MLCs of pure PMW require no anneal because the pre-existing B-site order is preserved by the low-temperature sintering (950 °C) that is sufficient for good densification. For our MLCs of PST–PMW, intermediate-temperature sintering (1,250 °C) is high enough for densification but low enough to preserve B-site order between high-valence and low-valence cations, thus avoiding the need to anneal.

Our values of $|\Delta T_{eff}|$ (Fig. 4) and efficiency (Fig. 5f) were obtained while driving EC effects in a repeatable way by using the same voltage (600 V) that is used to repeatedly drive geometrically similar MLCs of PST in prototypes[1–4]. This voltage was applied 644 times during our 322 bipolar cycles (Fig. 2) and more than $1.5 \times 10^7$ times during fatigue tests in which we observed no drop in EC performance (Methods).

Our work should lead to the replacement of MLCs of PST with MLCs of PST–PMW in EC prototypes, as cooling from above to below ambient is widely valued. Given that our EC effects at 600 V are not saturated (Supplementary Notes 9 and 15), processing improvements that increase breakdown voltage would yield even larger EC effects. Such improvements would be more effective in PST–PMW than PST, as anhysteretic single-phase EC effects exceed EC effects associated with the first-order transition by a greater amount (120–160% in PST–PMW, 23% in PST). More generally, our materials development strategy (valence-size sintering and ferroelectric–antiferroelectric solid solutioning) should inspire improvements in other ceramic materials, for EC applications and beyond.

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

# Article

## Methods

### Samples

The active area of MLC1 of 85PST–15PMW was used for STEM. MLC1 of 85PST–15PMW and MLC1 of 90PST–10PMW were used for indirect EC measurements from which efficiency was evaluated and for direct measurements of EC temperature change that were converted to effective temperature change. MLC2 of 85PST–15PMW and MLC2 of 90PST–10PMW were crushed to form powders for XRD. MLC2 of 85PST–15PMW and MLC2 of 90PST–10PMW were used for zero-field calorimetry and direct measurements of EC heat. MLC3 of 85PST–15PMW and MLC3 of 90PST–10PMW were used for optical images. MLC4 of 85PST–15PMW and MLC4 of 90PST–10PMW were used for dielectric spectroscopy. MLC5 of 85PST–15PMW was tested for fatigue. MLC6 of 85PST–15PMW was used to measure leakage current.

### MLC fabrication

Powders of high-purity $Pb_3O_4$, $Ta_2O_5$ and $MgWO_4$ (Kojundo Chemical Laboratory Co., Ltd) and $Sc_2O_3$ (Shin-Etsu Chemical Co., Ltd) were weighed, ball-milled for 16 h in distilled water with balls of partially stabilized zirconia, dried and then pulverized. The powder mixture was then calcinated at 850 °C for 4 h in air. Subsequent ball-milling that used partially stabilized zirconia balls with an organic solvent and binder resulted in a slurry for casting green sheets using the doctor blade technique. These green sheets were sliced, electroded by screen-printing Pt paste, stacked, pressed and cut to obtain green MLCs. After burning off the binder at 500 °C for 24 h, the proto-MLCs were sintered at 1,250 °C for 4 h in a Pb-containing atmosphere. Terminals were formed by painting on a bespoke Ag paste and firing in air at 750 °C for 10 min.

### MLC geometry

MLCs of 85PST–15PMW had external dimensions of $10.10 \times 7.18 \times 0.78$ mm³, an active volume of 32.6 mm³ that comprised 19 layers of thickness 35 µm and area 49 mm², Pt inner electrodes of thickness 2 µm and an active volume fraction of $v_{active} = 58\%$. MLCs of 90PST–10PMW had external dimensions of $10.20 \times 7.28 \times 0.78$ mm³, an active volume of 32.8 mm³ that comprised 19 layers of thickness 36 µm and area 48 mm², Pt inner electrodes of thickness 2 µm and $v_{active} = 57\%$.

### Thermalization of active and inactive layers

Within the active area and away from the periphery, we identify $|\Delta T_i| = f_i |\Delta T| = 0.85 |\Delta T|$ for both PST–PMW compositions by assuming complete thermalization between the 19 active PST–PMW layers (that would develop adiabatic temperature change $|\Delta T|$ if isolated) and the inactive layers that comprise two outer layers of PST–PMW and 20 inner Pt electrodes. The calculation assumes an off-peak volumetric heat capacity of $c \approx 2.5$ MJ K⁻¹ m⁻³ for PST–PMW at 273 K (Supplementary Note 4) and $c \approx 2.8$ MJ K⁻¹ m⁻³ for Pt.

### XRD

We used a Bruker D8 Advance diffractometer with Cu-Kα radiation and a LYNXEYE EX detector that obviates the need for a monochromator. For MLCs crushed to a powder, we obtained $2\theta–\omega$ step scans with a $2\theta$ step of 0.01°, a scan speed of 1.5 s step⁻¹ and a fixed illuminated length that increases the effective scattering volume at higher values of $2\theta$. The diffraction profiles we present and analyse were corrected with DIFFRAC.EVA software to make the effective scattering volume constant. Lattice parameters of $a_c \approx 8.13$ Å (85PST–15PMW) and $a_c \approx 8.14$ Å (90PST–10PMW) were estimated from the positions of the 422 reflections generated by Cu-Kα1 radiation (Supplementary Note 2). The intensities of the unsplit 111 and 200 reflections, determined by fitting pseudo-Voigt functions, were used to calculate the B-site order between high-valence and low-valence cations ($S_{111} \approx 0.99$ for 85PST–15PMW, $S_{111} \approx 0.97$ for 90PST–10PMW) (Supplementary Note 2).

### STEM

Following the protocol developed in ref. 43, a FEI Helios NanoLab DualBeam focused ion beam scanning electron microscope was used to prepare a cross-sectional lamella that was cut from the active volume of MLC1 of 85PST–15PMW and thinned below 30 nm for optimal atomic resolution. STEM characterization was performed at 300 kV on a Thermo Fisher Scientific Spectra 300 transmission electron microscope equipped with a high-energy-resolution extreme field-emission gun/mono monochromator. STEM-HAADF images were acquired using a beam current of 85 pA, a camera length of 58 mm, a convergence angle of 30 mrad, a dwell time of 5 µs and spatial sampling of 5 pm per pixel. Velox software was used to subtract backgrounds with a radial Wiener filter and generate intensity profiles.

### Differential scanning calorimetry

We identified the zero-field volumetric heat capacity $c(T)$ by experimentally determining $dQ/dT$ on heating and cooling in a TA Instruments Q2000 DSC with a sapphire reference. The relative zero-field entropy $S'(T) = S(T) - S(T_0) = \int_{T_0}^{T} c(T')/T' dT'$ is identified with respect to the entropy $S$ at $T_0 = 200$ K (85PST–15PMW) or $T_0 = 219$ K (90PST–10PMW). As shown in Supplementary Fig. 7, comparison of $S'(T)$ with respect to the corresponding function derived from a sigmoidal baseline for $c(T)$ yields thermally driven zero-field entropy changes of $|\Delta S_0| = 13.4$ kJ K⁻¹ m⁻³ (85PST–15PMW) and $|\Delta S_0| = 14.8$ kJ K⁻¹ m⁻³ (90PST–10PMW).

### Highly isothermal measurements of EC heat

We used a bespoke differential scanning calorimeter. The MLC under test and a copper reference were attached with thermal paste (RS-GCS-SP50, RS Components) to $10 \times 10$-mm sense and reference Peltiers that were monitored using a Keithley 2110 Multimeter but the copper reference is redundant for our isothermal measurements. The Peltiers were superglued to a massive copper block and a copper lid created a closed air space. The resulting assembly was placed in a steel cup whose temperature was set using near-total immersion in a LAUDA Eco Gold 1050 temperature bath. Cling film over the steel cup prevented ingress of the oily fluid. A Pt100 resistance thermometer, inserted into a channel in the copper block and monitored using a Keithley 2110 Multimeter, was used to identify sample temperature given that block temperature and electric field were only varied slowly. The measured temperature was stable to $|dT/dt| < 0.01$ K s⁻¹. Electric field was varied using a Keithley 2470 SourceMeter that was connected to the MLC under test using 0.2-mm-diameter coiled copper wires and silver paste. Specifically, we applied and removed $E = 21.4$ V µm⁻¹ at $dE/dt = \pm 7.1$ mV µm⁻¹ s⁻¹ (85PST–15PMW) and $E = 20.8$ V µm⁻¹ at $dE/dt = \pm 6.9$ mV µm⁻¹ s⁻¹ (90PST–10PMW), such that the duration of each unidirectional field sweep (approximately 3,000 s) was two orders of magnitude larger than the 10 s thermal relaxation time for 1/e decay. This thermal relaxation time is as large as it is because the heat flowing between MLC and copper block passes through the intervening Peltier that is used to measure it. The field ramps resulted in a differential heat $dQ/dt$ that was identified from the voltage difference measured between sense and reference Peltiers with the calibration given in ref. 41. After setting a repeatable unipolar branch by performing one of the field cycles described above, data were obtained on heating to temperatures that fell roughly every 4.9 K from 228 K to 356 K (85PST–15PMW) or roughly every 4.9 K from 238 K to 356 K (90PST–10PMW).

### Set-up for highly adiabatic EC measurements at different set temperatures

The variable-temperature calorimeter that we used for highly isothermal measurements of heat was used here but not as a calorimeter. Instead, the sense Peltier functioned as a passive extension of the copper block. An assembly comprising the MLC under study and

a thin copper plate, separated by two pieces of wooden toothpick on either side of the active area, was attached to the sense Peltier via the copper plate. The thus-mounted MLC showed a long thermal relaxation time of 33 s for 1/e decay. The MLC–toothpick connection was made using vacuum grease, the toothpick–copper plate connection was made using Loctite superglue (Henkel Adhesives) and the copper plate–Peltier connection was made using RS-GCS-SP50 thermal paste (RS Components).

### Highly adiabatic measurements of electrical polarization
A Keithley 2470 SourceMeter was used to acquire bipolar measurements of polarization in ±600 V by setting constant-magnitude currents (250 µA) that were large enough to set the duration of each unidirectional voltage sweep to <1.3 s, which is much smaller (<4%) than the 33 s thermal relaxation time. These measurements were acquired on heating to nearby values of set temperature that fell roughly every 0.45 K from 225 K to 370 K (85PST–15PMW) or roughly every 0.48 K from 228 K to 370 K (90PST–10PMW).

### Highly adiabatic measurements of EC temperature change
A Keithley 2470 SourceMeter was used to apply and later remove up to 750 V while limiting the magnitude of the current to a high value (10 mA). The resulting EC effects were driven during a field-change time of about 0.1 s. Temperature was measured using a K-type thermocouple (Therma Thermofühler GmbH) with junction diameter 400 µm and wire diameter 80 µm. The thermocouple was pressed onto the MLC face centre by means of its wires, held with a drop of black paint (Electrolube PNM400) and monitored at about 4.9 Hz using a Keithley 2110 Multimeter. After setting a repeatable unipolar branch by performing one of the voltage cycles described above, data were obtained on heating to set temperatures that fell approximately every 4.7 K from 235 K to 372 K (85PST–15PMW) and from 246 K to 376 K (90PST–10PMW).

### Highly adiabatic measurements of fatigue
Using a Radiant Precision Premier II with a Trek 609E-6 amplifier, we applied a 10-Hz triangular pulse of magnitude 600 V (17.1 V µm$^{-1}$) to MLC5 of 85PST–15PMW at a set temperature near which EC effects peak (264 K; Fig. 3d). Under this driving protocol, $5 \times 10^6$ bipolar cycles were followed by $5 \times 10^6$ unipolar cycles, such that 600 V was applied $1.5 \times 10^7$ times. During a limited number of interruptions to the $1.5 \times 10^7$ cycles, and after all $1.5 \times 10^7$ cycles were complete, field-on and field-off measurements of $|\Delta T_j|$ with $|E| = 17.1$ V µm$^{-1}$ lay within 4% of the corresponding values measured at the outset.

### Dielectric permittivity
Data for Supplementary Note 5 were measured using an Agilent 4294A analyser that was electrically connected to the MLC under test by means of two W needles that also served to mechanically connect the MLC via an electrically insulating glass coverslip and underlying thermal paste to a variable-temperature Linkam stage, which was heated and subsequently cooled at ±0.083 K s$^{-1}$. MLC temperature was identified using a Keithley 2110 Multimeter to record the temperature of a thermocouple attached to the MLC face centre using a drop of black paint.

### Steady-state high-field power dissipation
We employed two methods that yielded same order-of-magnitude values of power. While applying 21.4 V µm$^{-1}$ to MLC2 of 85PST–15PMW at 297 K, the dissipated power was 8.15 µW, as deduced from our sensitive calorimetry (Supplementary Fig. 16c). While applying the same field to MLC6 of 85PST–15PMW at 295 K, the dissipated power was 30.3 µW, as deduced using a Keithley 2470 SourceMeter to measure for 1 h a steady-state leakage current of 40.4 nA (43.4 µA m$^{-2}$). The resistance corresponding to this current is 18.6 GΩ ($4.93 \times 10^7$ µΩ cm).

### Identification of the phase boundary
Inflection points in $P(E)$ at measured values of set temperature $T_s$ were plotted on $(S', E)$ axes (Fig. 2a–d) after converting $T_s$ to $S'$ (main-text discussion of Fig. 2a) and smoothing the resulting phase boundary with a Savitzky–Golay filter. The phase boundary in $T(S', E)$ (Fig. 2d) gives the relationship between $T$ and $E$ that we use to identify the phase boundary on $(T, E)$ axes (Fig. 2e,f).

## Data availability
The authors declare that the data supporting the findings of this study are available in the paper and its Supplementary Information file. Data for PST and Gd are available at https://doi.org/10.1038/s41586-019-1634-0 (ref. 5). Source data are provided with this paper.

43. Schaffer, M., Schaffer, B. & Ramasse, Q. Sample preparation for atomic-resolution STEM at low voltages by FIB. *Ultramicroscopy* **114**, 62–71 (2012).

**Acknowledgements** We thank K. Sasaki and H. Kuramoto for their assistance in fabricating the MLCs. We acknowledge the use of the University of Cambridge Wolfson Electron Microscopy Suite and the use of the Thermo Fisher Scientific Spectra 300 that was financed under UK EPSRC grant EP/R008779/1. M.G. was supported by a Newton International Fellowship from the Royal Society and a Goldsmiths' Early Career Research Fellowship from Churchill College, Cambridge. X.M. acknowledges support from UK EPSRC grant EP/M003752/1, ERC Starting grant 680032 and the Royal Society. V.F. was supported by St. John's College, Cambridge. Y.T. and J.Z. were supported by CSC Cambridge scholarships from the China Scholarship Council and the Cambridge Trust. A.Z.K.G. was supported by the EPSRC Cambridge NanoDTC grant EP/S022953/1. We thank J. Harada, E. Defay, T. Usui and G. G. Guzmán-Verri for discussions.

**Author contributions** M.G., X.M., S.H. and N.D.M. led the project. S.H. was responsible for fabricating the MLCs. M.G., Y.T. and M.V. were responsible for the XRD experiments. X.C., S.M.F. and C.D. were responsible for planning, executing and interpreting the STEM experiments. M.G. and V.F. improved the hardware and software for the electrical and thermal measurements. M.G. performed all electrical and thermal measurements, with support from V.F., A.M., A.Z.K.G. and J.Z. M.G. curated the selection of STEM data and either analysed all other data or supervised their analysis by A.M. and A.Z.K.G. N.D.M. wrote the manuscript and supplementary information file with M.G., using substantive feedback from X.M. and using further feedback from S.H. and C.D., except X.C. wrote the STEM methods using feedback from C.D.

**Competing interests** S.H. has filed a provisional patent on PST–PMW materials (WO2023/190437). The other authors declare no competing interests.

**Additional information**
**Correspondence and requests for materials** should be addressed to M. Guo, X. Moya, S. Hirose or N. D. Mathur.

**Extended Data Table 1 | Indirectly measured peak EC effects for MLCs of PST-PMW with 600 V**

| Composition | $|\Delta T|$ (K) | $|Q|$ (MJ m$^{-3}$) | $|\Delta S|$ (kJ K$^{-1}$ m$^{-3}$) |
|---|---|---|---|
| 85PST-15PMW | $4.1_{279\,K}$ | $9.5_{287\,K}$ | $34.6_{268\,K}$ |
| 90PST-10PMW | $3.9_{282K}$ | $9.3_{285\,K}$ | $32.9_{281\,K}$ |

Peak values occur at subscript temperatures. For 85PST-15PMW, data are from Fig. 2c,f, $E$ = 17.1 V µm$^{-1}$. For 90PST-10PMW, data are from Supplementary Fig. 10c,f, $E$ = 16.7 V µm$^{-1}$.

**Extended Data Table 2 | Directly measured peak EC effects for MLCs of PST-PMW with 750 V**

| Composition | $|\Delta T_i|$ (K) | $|Q|$ (MJ m$^{-3}$) |
|---|---|---|
| 85PST-15PMW | 3.3 $_{283\,K}$ | 10.5 $_{292\,K}$ |
| 90PST-10PMW | 3.2 $_{288\,K}$ | 10.4 $_{292\,K}$ |

Peak values occur at subscript temperatures. For 85PST-15PMW, data are from Fig. 3b,d, $E$ = 21.4 V μm$^{-1}$. For 90PST-10PMW, data are from Supplementary Fig. 15b,d, $E$ = 20.8 V μm$^{-1}$.