## [Peer Review File · Nature]

Electrocaloric effects across room temperature in multilayer capacitors

Corresponding Author: Dr Mengfan Guo

Version 0:

Reviewer comments:

Referee #1

(Remarks to the Author)

The authors present a promising MLC based on solid solutions of PST and antiferroelectric PMW, which demonstrates large EC effects above and well below room temperature. Compared with PST MLCs prototypes, the PST-PMW MLCs reported in this work show impressively low Curie temperature (~230 K), addressing application challenges of EC effects in low temperatures. Overall, this study presents excellent research findings and can be accepted with a few revisions to the manuscript and the Supporting Information.

Major Comments:

1- To help readers better evaluate the practical cooling capability of the reported MLCs, I recommend including the thermal conductivity of the PST-PMW, either as measured or from reasonable estimates. Additionally, a brief calculation converting the measured temperature change ($\Delta T \sim 3$ K) into an estimated heat flux at a typical EC system operation frequency would provide meaningful context. While the EC temperature shift reflects intrinsic material performance, the actual heat flux also depends on how quickly heat can be absorbed or dissipated from the active region. Including this analysis would add value from a device engineering standpoint.

2- While the authors describe a direct measurement of EC temperature change using a thermocouple affixed to the MLC surface, several key experimental details remain unclear. Given the importance of this measurement in validating the EC performance, it would strengthen the manuscript to provide additional information on the thermocouple type, response time, the thermal contact quality between the thermocouple and the sample (e.g., thermal interface resistance introduced by the black paint? The black paint thermal conductivity?), and the thermal environment during measurement. For instance, was the sample mounted under thermal insulation to minimize heat loss? Additionally, while the measurements were conducted down to 240 K, the manuscript does not specify how the low-temperature environment was achieved or stabilized. Clarifying the cooling method and temperature control accuracy would enhance the credibility and reproducibility of the results.

Minor suggestion :

- 1 . "CrossleyPRX" appears on page 5.
- 2 . Check the consistency of the formatting for T_c on page 8.
- 3 . Include legends for the curves in Figs. 3 c-d and S10 c-d to enhance readability.
4. The language could be further refined by avoiding subjective statements like "We see" and "Let us now compare" in the discussion.
5. The title of the SI is different from that of the main text.

Referee #2

(Remarks to the Author)

EC effects in ferroelectrics have attracted growing interest for cooling applications. The authors studied EC effect in PST-PMW multilayer capacitors which shows the existence of large EC responses above and below room temperature. Based on their previous works using PST (Nature 575, 468 (2019) or ref. 5), they introduce PMW to form solid solutions. They show that the phase transition temperature is markedly lower from 290 K to 230 K. As a result, large EC effects occur at lower

temperatures, expanding the operating temperature window which has not been accessible in their previous Nature work (Nature 575, 468 (2019) or ref. 5) as their data were limited to above room temperature. Meanwhile, the efficiency of PST-PMW multilayer capacitors remains slightly larger than PST. With a comparable EC response and efficiency, PST-PMW requires no long-term annealing like PST and exhibits a wider working temperature window which offers a promising alternative to replace PST used as the key elements for design of EC prototype devices. Overall, the work on PST-PMW represents advances over PST according to EC data measured at different temperatures and electric fields through both direct and indirect methods.

Despite these improvements to the previous Nature work published in 2019, the results obtained in the current work fail to demonstrate significant advances compared with the state-of-the-art studies including modified PST ceramics, lead-free and lead-contained EC multilayer capacitors, and previous results on multilayer capacitors based on PST solutions. Moreover, the manuscript only focused on comparison between PST-PMW and PST (Nature 575, 468 (2019) or ref. 5) by still keeping the data on PST already published without expanding them to lower temperature at least for better comparison. As one can see from Fig. 4, one may expect a better EC efficiency and possibly a wider working temperature window for PST alone. Besides these PST-based systems, the state-of-the-art results are not considered. Indeed, there have been several recent publications reporting the EC cooling near and below room temperature which is comparable or larger than the results obtained in this manuscript. Therefore, the main novelty claiming large electrocaloric effects above and below room temperature is significantly diluted. Note that the same group has already published EC responses, while not as high, below room temperature in another system (Nature Materials 23, 639 (2024)) and there are already published works showing high EC properties below room temperature. The consideration of the existing literature, in addition to PST-based systems, must be also considered. Moreover, this work lacks sufficient fundamental insights into the phase diagram of PST-PMW including composition, temperature and electric field with emergence of relaxor behavior which is essential to design large EC response. In particular, the local structures related to relaxor phase caused by PMW and the correlations with EC response remain unknown. In addition, other technical issues such as the conditions for calculating efficiency and the comparison between PST-PMW and Gd rather than state-of-the-art magnetocaloric materials remain yet to be addressed. Therefore, based on these considerations, the reviewer cannot recommend the publication of the current manuscript in Nature. In the following, please find some comments and concerns the authors may consider if they wish to submit their work somewhere else.

#1 The authors wrote “Consequently, a growing number of prototype EC coolers that exploit assemblies of thin ceramic/polymer layers (MLCs)^{1-4,15-24} and polymer bilayers²⁵⁻²⁹/monolayers^{30,31} can cool loads down to temperatures that lie near room temperature. However, EC cooling through room temperature is yet to be achieved, even though such cooling is important for food, beverages, the built environment, and medicine, etc.”

Here the central comment on previous studies raised by the authors is that previous works mainly focused on the working temperature range near or above room temperature. Moreover, the main novelty based on this argument is that they have achieved EC cooling through room temperature.

This argument is not fully true as a few recent works reported large EC properties through room temperature in various EC materials. For instance, a very recent work (J Adv Ceram 2025, 14: 9221088. <https://doi.org/10.26599/JAC.2025.9221088>) entitled “Optimizing electrocaloric effect of PbSc_{0.5}Ta_{0.5}O₃ ceramics near/below room temperature by ordering degree modulation” reported that through tuning of ordering degree the working temperature range is lowered near and below room temperature. It reported an EC-induced temperature change of 1.5 K under an electric field of 6.0 V/μm at -10°C (or 263 K) for $\omega = 0.51$ which is comparable that obtained in PST-PMW under the same temperature and field. Meanwhile, this recent work used lattice ordering to tune the working temperature range for achieving large EC effect in PST ceramics from 40°C (or 313 K) to -10°C (263 K). Besides, the fabrication of modified PST ceramics does not require long-term annealing as required in previous Nature work (Ref. 5). Therefore, the current manuscript shows the absence of striking advances over this recently published work as they both show the same level EC response, and both involve the operating temperature below room temperature. For instance, for modified PST, ordering degree was used to shift phase transition temperature towards lower temperatures through room temperature while for PST-PMW, incorporation of PMW lowers the phase transition below room temperature. This strongly challenges the argument by the authors that “EC cooling through room temperature is yet to be achieved” and “an energetically expensive 42-day anneal is required to achieve good B-site order”.

#2 The authors wrote “For example, indirect measurements show that a thin plate of B-site disordered PST exhibits an adiabatic temperature change of magnitude $|\Delta T| = 0.6$ K at 260 K, while a thin plate of PST-PbSc_{0.5}Sb_{0.5}O₃ exhibits $|\Delta T| = 1.5$ K at 276 K.”

In this sentence, the authors recognized that the phase transition temperature of PST has been lowered below room temperature through solid solution method (Ferroelectrics 127, 143 (1992) or ref. 32). Moreover, the authors overlooked the electric field used to trigger the large EC-induced temperature change of 1.5 K at 276 K. It is found that the electric field used in this previous work (Ferroelectrics 127, 143 (1992) or ref. 32) was 25 kV/cm as identical as 2.5 V/μm. This field is much lower than that used in current work. For instance, under the same electric field of 2.5 V/μm, the temperature change is about 0.8 K which is less than half of that reported in previous work (Ferroelectrics 127, 143 (1992) or ref. 32). Meanwhile, the operating temperature for PST-PbSc_{0.5}Sb_{0.5}O₃ is from -20°C to 10°C which also covers the temperature through room temperature, in contrast with the claim that “However, EC cooling through room temperature is yet to be achieved”. Both studies use the solid solution method to mainly introduce modification on B-site: the previous one used PST-PbSc_{0.5}Sb_{0.5}O₃ while this work uses PST-PMW. The smaller EC strength in current work than previous one strongly declines the advance of current work given that EC strength of EC materials is crucial to practical cooling applications.

#3 PMW is known as an antiferroelectric and EC studies in PMW were conducted by several groups. For instance, PMW including both bulk ceramics (Adv. Funct. Mater. 2021, 31, 2101176) and multilayer capacitors (Adv. Funct. Mater. 2025, 2502550 DOI:10.1002/adfm.202502550) showing large EC temperature change have been studied. To make a direct

comparison, here we mainly focus on PMW multilayer capacitors under the same electric field of about 21 V/ μm . This previous work shows a large temperature change of 4.0 K at 100°C (or 283 K) which remains 2.9 K at -100°C (or 263 K). This add another example challenging the authors' claim that "However, EC cooling through room temperature is yet to be achieved". For comparison, in the present manuscript, PST-PMW shows a temperature change of about 3.3 K at 283 K and about 3.2 K at 263 K driven by an electric field of 21.4 V/ μm . These results indicate that the EC effects obtained under the same electric field and temperatures in PST-PMW are comparable to those obtained in PMW multilayer capacitors published recently (Adv. Funct. Mater. 2025, 2502550 DOI:10.1002/adfm.202502550).

Moreover, PMW at near room temperature shows a larger temperature change of about 6.7 K at 293 K which is twice as large as that of PST-PMW in the present work. Also note that as PMW is antiferroelectric, it exhibits a combination of both negative (converse) and positive (direct) EC response depending on the working temperature. In previous work (Adv. Funct. Mater. 2025, 2502550 DOI:10.1002/adfm.202502550), it is demonstrated that COP of PMW multilayer capacitors is much higher than PST counterparts. As efficiency of PST-PMW multilayer capacitors is only slightly larger than PST multilayer capacitors, COP of PST-PMW multilayer capacitors is nearly the same as that of PST multilayer capacitors. Therefore, PMW multilayer capacitors may compete more favorably than complex PST-PMW multilayer capacitors, as already it shows large EC effect through room temperature and do not need long-time annealing. This again challenges the authors' argument in the introduction part "However, EC cooling through room temperature is yet to be achieved, even though such cooling is important for food, beverages, the built environment, and medicine, etc."

#4 It is suggested to add the temperature-composition-electric field phase diagram of PST-PMW which is crucial to understanding the EC behavior in PST-PMW. Huge efforts were made by the authors in characterizing the EC properties. However, fundamental insights allowing to inspire further material design remains lacking in the current version of manuscript. As mentioned in comment #3, PMW is antiferroelectric differing from ferroelectric-like PST with ordered B-site cation order. How the symmetry phase, state, and phase transition evolve with incorporation of PMW—that is crucial to EC response—remains elusive based on the current data. Temperature-dependent characterization of phase transition behavior based on XRD and dielectric spectra is required. For instance, the temperature dielectric spectra can be used in complementary to XRD or other techniques to analyze the phases and the critical temperatures. In this regard, only three frequencies were shown in Fig. S14, which are not enough to analyze the relaxor behavior. In addition, it is shown in Fig. S14 that in 85PST-15PMW it seems that there is another transition, given the kink on the left side of the dielectric peak maximum. The losses could also help the authors to identify the real behavior of the dielectric response here. There is also a strange behavior (could be related to defects) related to the frequency changes of the maximum of the dielectric constant when both 90PST-10PMW and 85PST-15PMW are compared. The abnormal/strange dynamic dielectric behavior here stays unclear and require more efforts, not mentioning the effect of applied dc-electric field.

#5 The local structure of PST-PMW with inclusion of PMW remains elusive based on current data. For instance, the authors mentioned that "Good B-site cation order in 85PST-15PMW is confirmed via HAADF-STEM". However, HAADF-STEM data shown in Fig. 1d and Fig. S3 are of low quality compared with previous work on PMW (Adv. Funct. Mater. 2021, 31, 2101176; Adv. Funct. Mater. 2025, 2502550 DOI:10.1002/adfm.202502550). It is suggested that STEM with high quality image should be provided. In addition, the authors recognized the presence of local chemical disorder by saying that "B2 columns comprise the high-Z elements (Ta⁵⁺ with Z = 73, W⁶⁺ with Z = 74) and that the B1 columns comprise the low-Z elements (Sc³⁺ with Z = 21, Mg²⁺ with Z = 12)". Meanwhile, temperature-dependent dielectric spectra show the existence of relaxor behavior. How do the local chemical disorders inherent to relaxor behavior correlate with EC response? This is also critical to understanding a very broad EC response with respect to temperature.

#6 The cycle efficiency. The use of cycle efficiency might potentially mislead the readers as the number as large as 70-90% is calculated under ideal conditions. Also, the word "prototype" should be indicated specifically to readers as multilayer capacitors are the key elements used in EC cooling prototype devices. Therefore, strictly speaking, multilayer capacitors alone cannot fully represent the prototype devices. Here the reviewer's point is that the performance of real prototype devices based on multilayer capacitors is more convincing to demonstrate the high-efficient cooling applications. Indeed, a lot of EC device publications claimed high COP or efficiency while they did not specially show readers the conditions how high COP or efficiency is achieved. For instance, an Erratum for a recent Science paper (<https://www.science.org/doi/10.1126/science.adz0508>) say that "Hence, the maximum coefficient of performance of the electrocaloric heat pump—which is affected both by the cooling power and the recoverable energy—reaches 54% of Carnot's efficiency if 99% of the recoverable energy is harvested." Like the efficiency calculated here, cooling power based on PST multilayer capacitors is calculated under the condition only if 99% of the recoverable energy is harvested. Unfortunately, readers are unaware of the real percentage of recoverable energy under the real conditions. As these numbers are calculated based on equations rather than direct measurements on real prototype devices, using incorrect equations may yield overestimation of true values, misleading the readers and the field. More specifically, the authors assume work recovery by following the ref. 21. More information should be provided regarding work recovery. What is the real value of the work recovery in real devices? Please provide detailed information on how to technically testify the work recovery and its role in affecting efficiency to avoid potential overestimation of the device performance.

#7 Usually, it is believed that efficiency and COP is positively dependent on EC response (Nature Materials 13 (5), 439-450 (2014); Nature 600 (7890), 664-669 (2021)). This is why large EC effect under low fields is required for the design of cooling devices. However, the authors show that "This increase of efficiency arises due to the small concomitant reduction in the magnitude of the EC effect (Fig. 4)". More explanation should be provided on enhanced efficiency while reducing EC response.

#8 In previous Nature work (Nature 575, 468 (2019) or ref. 5), the authors have made a comparison of EC effect with

magnetocaloric effect in Gd under different temperatures to demonstrate the advance of EC multilayer capacitors for solid-state cooling applications. Six years later, the results on Gd are still used for comparison. Note that magnetocaloric effect in Gd has been well studied since 1976 (G. V. Brown. J. Appl. Phys., 47,3673, (1976)). There have been many advances in magnetocaloric materials, especially during the past two decades which show better magnetocaloric response than Gd (Progress in Materials Science 93, 2018, 112-232). Many prototype devices have been fabricated based on various magnetocaloric materials (Kitanovski, A. et al. (2015). Overview of Existing Magnetocaloric Prototype Devices. In: Magnetocaloric Energy Conversion. Green Energy and Technology.). It is therefore suggested to add the comparison with the state-of-the-art magnetocaloric response instead of Gd.

#9 "Like MLCs of PST (ref. 5), supercritical fields yield large EC effects over a wide range of temperatures". Supercritical phase transition has been used in PST reported in previous work (ref. 5) to rationalize the large EC effect in PST under high fields. What is the unique feature in PST-PMW whereas PMW may tune the supercritical phase transition behavior. If there is not, the inclusion of PMW shifts the transition temperature towards lower temperature. As PST and PMW have both extensively studied during the past 6 years, the combination of PST and PMW is not considered as a breakthrough to the field even though EC effect has not been studied in PST-PMW. This lacks sufficient conceptual advance as required by the high standard in Nature.

#10 The partial substitution disrupts dipolar order while preserving crystallographic order, resulting in first-order transitions with reduced Curie temperatures of $T_C \sim 230$ K (85PST-15PMW) and $T_C \sim 242$ K (90PST-10PMW) measured on heating (Supplementary Note 4). The thermally driven zero-field entropy change for the transition of $|\Delta S_0| = 13.4$ kJ K⁻¹ m⁻³ in 85PST-15PMW which is smaller than $|\Delta S_0| = 14.8$ kJ K⁻¹ m⁻³ obtained in 90PST-10PMW. This indicates the smearing of the phase transition near T_C . However, the thermal hysteresis increases from 10 K to 14 K as PMW concentration increases from 10 PMW to 15 PMW. What is the origin of increased thermal hysteresis here?

#11 PST or PST-PMW contain toxic lead while a recent work on lead-free high-polar-entropy ceramics exhibits higher EC-induced temperature change near room temperature (Nature 640, 924930 (2025)). For instance, an EC temperature change of 10 K is achieved at an electric field of 10 V/ μ m near room temperature which is 3 times as large as that obtained by lead-contained PST-PMW under a higher electric field of 21.4 V/ μ m. EC response below room temperature was also studied in this previous work (Nature 640, 924930 (2025)). Moreover, MLCs based on lead-free ceramics have also been fabricated (Nature 640, 924930 (2025)). It is suggested to add a direct comparison of the magnitude of EC response near room temperature between current work and previous results (Nature 640, 924930 (2025)) to address the main advance over this recent work in Nature.

Referee #3

(Remarks to the Author)

Key results: Guo et al. report on a solid solution of PST and PMW which show electrocaloric effects on the order of several Kelvin over a broad range of temperature, around, and including room temperature. The MLCCs are robust against field fatigue ($>10^7$ cycles) and can in principle show high efficiencies (70-90%) when incorporated into regenerator coolers. This new composition significantly improves the temperature span of regenerator coolers, compared to PST, while also being more straightforward and less energetically costly to manufacture (long annealing steps required for PST are obviated). Comprehensive electrical and thermal characterization of the samples has been carried out to benchmark performance with good agreement between indirect electrical and direct temperature measurements being obtained, therefore supporting the authors' conclusions about the electrocaloric performance of PST-PMW MLCCs.

Originality and significance: The fact that this new composition significantly improves the temperature span of regenerator coolers while also being more straightforward and less energetically costly to manufacture (long annealing steps required for PST are obviated) represents a substantial step forward for electrocalorics research. PST has been considered to be the best electrocaloric material and PST MLCCs have been used in the leading electrocaloric cooler prototypes (Wang et al. Science 370 370, 129 2020; Li et al. Science 382, 801, 2023; Metzdorf et al. Commun. Eng. 3, 55, 2024). While there is a trade off in diluting PST with PMW in terms of magnitude of EC effect, this is made up for by the increased operating span across room temperature, making it very promising for applications. As the author's point out, there is still room for further optimization based on the strategy of valence-size sintering that has been outlined here.

Clarity and context: the text is well written, succinct and clear to read. I think it is accessible for a generalist audience and communicates the important points well. The references cited are suitable and provide a fair reflection of the state of the art.

Queries / Suggested improvements:

The authors outline their choice for choosing PMW to alloy with PST on the basis of comparable B-site cation dissimilarity and larger valence disparity. Since this design principle will be of great interest to readers, it would be useful for readers if they could suggest other candidate solid solutions that might be worth exploring.

How have the authors decided upon the chosen solid solution compositions of 10% and 15% PMW? This does not seem to be discussed in the paper or a framework for how the proportion of dilution by PMW is expected to tune the electrocaloric response and temperature span. It would be interesting to hear the authors' expectation for this and if they have been prevented from further exploration of the compositional phase space due to practicalities or challenges faced with sample fabrication

The authors suggest a relaxor-like behavior for PST-PMW supported by dielectric dispersion and well-established behavior for PST. Do the authors have any idea about the microstructure/polar configuration in the relaxor-like phase and how this is affected by PMW, which prefers antiferroelectric ordering (Li et al Adv. Func. Mater. 31, 2101176, 2021)? More of an explanation of the microscopic basis for the order-disorder entropy changes underpinning the observed EC effects would be useful or some clarification if the authors expect the effects to be primarily derived from latent heat as in the report for PST (Nair et al Nature 575, 468, 2019). In this regard, is there scope for discussion on the relative role of compositional disorder (B site cation distribution) versus dipolar entropy contributions, such as in the case of high polar entropy perovskites? (e.g. Du et al Nature 620, 924, 2025).

The robustness against fatigue over the course of 10^7 cycles at 600 V is impressive but have the authors determined what is the actual breakdown field for these samples and how it compares to other high performance electrocalorics prepared in MLCC format? If the breakdown field is superior to PST then there is perhaps scope to mitigate the lower EC temperatures achieved by driving the MLCC with larger fields.

In previous works the authors have used IR camera imaging to directly map temperature fields but here they have used only thermocouple measurement. I appreciate this is challenging for below room temperature but could be still be used for independent room temperature direct measurement verification. This would add further confidence to the reported EC temperature changes and reveal the spatial distribution of temperature across the MLCC devices.

Referee #4

(Remarks to the Author)

I co-reviewed this manuscript with one of the reviewers who provided the listed reports.

Referee #5

(Remarks to the Author)

I co-reviewed this manuscript with one of the reviewers who provided the listed reports.

Version 1:

Reviewer comments:

Referee #1

(Remarks to the Author)

The questions and concerns have been well addressed. Given the significance of the findings and the solid theoretical research, we are pleased to recommend this work for publication in Nature. We believe that it will make a significant contribution to the field.

Referee #2

(Remarks to the Author)

The referee would like to thank the authors for their efforts in revising the manuscript and for addressing the previous comments. However, the additional arguments/corrections don't mitigate the concerns raised by referee 2 including the novelty of the work, the involved physical mechanism or the practical breakthrough of the current work, even after revision. Besides, this manuscript also contains various inconsistencies between the measured EC responses and the theoretical understandings. Therefore, the referee 2 cannot recommend this work for publication in Nature.

First, the claim that "EC effects in existing material systems cannot be exploited for cooling through room temperature." is not fully correct as there are existing materials that have been developed for cooling through room temperature. To make a rational comparison of the EC properties, it is always useful to show the data deduced under the same electric field. This is crucial especially for the fact that PST-PMW layer in MLCC is much thinner than that in bulk ceramics which allows much higher electric field to be applied to drive higher EC response. In this regard, the referee mention here that they have for instance cited the modified PST ceramics reported in (R. Yin et al., J. Adv. Ceram. 14, 9221088 (2025)) which finally exhibits slightly higher EC-induced effect than that of PST-PMW (proposed in the current work) when considering the same electric field (i.e. 6 V/ μm). Meanwhile, the authors also recognized that PST-based solution exhibit comparable EC response to current work under the same electric field according to "Correcting 1.8 K for layer thermalization using the factor of 0.85 that we give in the main paper implies an effective temperature change of $|\Delta T_{\text{eff}}| \sim 1.5$ K, which is similar to the value quoted by the referee." It is not fully fair to impose the comment that these results are "below 1.6 K". The reason why PST-PMW achieves higher EC response than 1.6 K is the much higher electric field used in MLCC thanks to thinner thicknesses. As a matter of fact, ECE deduced from PST-PMW is not enhanced compared with PST solutions reported 33 years ago (Ferroelectrics 127, 143 (1992)) and state-of-the-art results in modified PST (J. Adv. Ceram. 14, 9221088 (2025)) without the need for long-term annealing and/or complex composition.

In our previous comment, the referee mentioned that "The smaller EC strength in current work than previous one strongly declines the advance of current work given that EC strength of EC materials is crucial to practical cooling applications." The authors further admitted that the EC strength is not enhanced through PST-PMW solid solution according to their sentence "After correcting the referee's estimate of 0.8 K to 1.5 K two answers ago, one finds that the two EC strengths are

equivalent.” This is also recognized by reviewer 3 that expanding the operating below room temperature is achieved at the expense of reduced EC response and EC strength. However, the authors said that “EC strength is of secondary importance and does not affect our main claim to have demonstrated an EC working body that could be used to cool a load through room temperature.” As the referee already mentioned in their previous review, this work does not represent the first example demonstrating “EC working body that could be used to cool a load through room temperature”. Please refer to the recent work on EC cooling through room temperature based on PMW MLCC (R. Yin et al., *Adv. Funct. Mater.* 2502550 (2025)) published before the submission of current work.

Second, although the authors tried to comment on previous works, the significant challenge why the previous works failed to achieve “EC effects exploited for cooling through room temperature” remains not clear to readers. Especially, the referee sees no key scientific mechanism preventing the practical use of EC ceramics and MLCC from cooling through room temperature that has been successfully overcome by the authors. From material’s perspective, EC effect below room temperature has been demonstrated with comparable EC value 33 years ago (*Ferroelectrics* 127, 143 (1992)). Indeed, the authors also recognized that “We are not claiming that the solid-solution method is novel, but please note that solid solutions of PST and PMW are “hitherto unexplored”.” PMW with larger enthalpy change than PST exhibit outstanding EC properties and cooling performance (COP, heat flux) is only mentioned with a short comment “decay to zero and change sign on crossing room temperature (as seen for antiferroelectric PMW)35-37”.

In addition, the mechanism for achieving EC response below room temperature proposed by the authors is that they “supercritically drive a first order ferroelectric phase transition above a low Curie temperature”, which is not new. Both supercritical phase transition and tuning of phase transition temperature using solution method were already proposed in the field, as also recognized by authors. Moreover, both EC response and EC strength is considerably reduced in the PST-PMW samples of the authors due to including PMW. Consequently, the EC results deducted from PST-PMW only make incremental contribution to the field at the material level while outstanding scientific understanding is not revealed in the current work. In addition, even the current mechanism to explain EC behavior is questionable as the referee already found the existence of serious inconsistencies in characterizing the phase transition which fails to provide understanding of observed EC properties.

Given the lack of sufficient conceptual breakthroughs, to meet the high standard required by Nature Journal, there must be striking advance for practical application in this work compared with previous studies. Unfortunately, the referee has already stated that the results obtained in PST-PMW are not sufficiently striking compared with the state-of-the-art MLCC based on PMW (R. Yin et al., *Adv. Funct. Mater.* 2502550 (2025)). The referee’s point is that both PST-PMW and PMW have advantages and disadvantages which are summarized as follows. PST-PMW shows broad temperature span while the counterpart of PMW is narrow; The magnitude of largest EC response in PMW considerably outperforms that of PST-PMW under the same electric field, corresponding to higher EC strength; Processing of PST-PMW is more complex and costly than PMW simply due to complex solid solution; Ag/Pd electrode was used in PMW while much more expensive Pt electrode (the price is about 3 times as that of Ag and roughly the same as Pd) was used; PMW uses an oscillation circuit designed to recover energy while PST-PMW simply assumes 100% energy recovery without any circuits; COP based on PMW is 350 while COP based PST-PMW is about 100; The heat flux of PMW is 245 W cm⁻² while it is about 1.4 W cm⁻² for PST-PMW (77.7 mW; area 5.6 mm²); The driving voltage for PMW is 350 V while it is 600 V for PST-PMW. According to the summary above, there is no key evidence supporting that PST-PMW is more favorable than PMW except the broader temperature span achieved in PST-PMW compared with PMW. In other words, PST-PMW fails to demonstrate striking advance over PMW for practical cooling applications. Even the authors recognized the promise of PMW in their previous paper that “This good EC performance near room temperature implies that MLCs of PMW could be exploited in prototype EC coolers.” (Sakyo Hirose et al 2023 *J. Phys. Energy* 5 035009). In addition, PST-PMW exhibits a cooling power of 77.7 mW assuming 100% energy recovery which is smaller than that (90 mW) obtained in real prototype based on PST MLCC (Y. Wang et al *Science* 2020, 370, 129) weakening the claim that “MLCs of PST-PMW should now replace MLCs of PST”. Nevertheless, the authors have added a new comment to distinguish the current work from previous works on PMW that “decay to zero and change sign on crossing room temperature (as seen for antiferroelectric PMW)35-37”. This comment is not fully acceptable as explained below. The former “decay to zero” is not relevant to EC cooling using PMW below room temperature as EC response in PMW only decays remarkably to zero only below -20°C. For practical cooling applications, -20°C is enough to meet the operating temperature range for cooling food, beverages, the built environment, and medicine as mentioned in the Introduction part. On the other hand, the sign change in EC response is not a drawback preventing PMW from practical applications. As this point has also been recognized in their previous paper (Sakyo Hirose et al 2023 *J. Phys. Energy* 5 035009) saying that “Note that we do not follow the common practice of describing inverse EC effects as negative EC effects because absolute sign depends on whether field is applied or removed, and whether one considers a temperature change in the adiabatic limit or an entropy change in the isothermal limit.”. The sign change can be simply overcome by applying electric field for the temperature range with a negative sign while removing electric field for the temperature range with a positive sign. Meanwhile, the sign change is not an issue if PMW is used for a specific temperature range near room temperature from 20°C to 35°C (the built environment) or 0-10 °C or -20-0°C (food, beverages, and medicine). Therefore, these analyses strongly suggest that ““decay to zero and change sign on crossing room temperature (as seen for antiferroelectric PMW)35-37”” cannot be regarded as a key factor hampering PMW for cooling through room temperature not to mention that current authors also supported the use of PMW for practical cooling applications (Sakyo Hirose et al 2023 *J. Phys. Energy* 5 035009).

For further evaluation of practical cooling applications, EC cooling cycles under ideal conditions have been studied by assuming 100% energy recovery. The authors referred to the work reported in ref. 44 whereas 99.74% energy recovery is achieved in PMN combined with specifically designed electrical circuits. Note that COP is strongly dependent on the value of the energy recovery during the cooling cycles as also recognized by authors in their previous paper (*Nature Communications* 9, 1827 (2018)).

Despite 99.74% energy recovery achieved in PMN, previous work by authors shows that 70-80% energy recovery is

achieved in BTO-based MLCC which decreases to 65% for a prototype refrigerator with 24 such capacitors (Nature Communications 9, 1827 (2018)). Without explicit demonstration using electrical circuits, the use of 100% energy recovery in PST-PMW MLCC remains yet to be technically testified simply because the electrical circuits were not really done in current work. Otherwise, one can argue that modified PST (R. Yin et al., J. Adv. Ceram. 14, 9221088 (2025)) should now replace MLCs of PST in EC prototypes for efficient EC cooling through ambient even though modified PST ceramics were not made into MLCC as done in this work.

The value of COP under ideal condition is about 100 according to Fig. 5e. By contrast, an oscillation circuit was designed to effectively recover the electric energy based on PMW MLCC (R. Yin et al., Adv. Funct. Mater. 2502550 (2025)). The COP with an oscillation circuit is 350 which is nearly 3 times higher than that obtained by PST-PMW without any circuit to recover the energy. This result explicitly demonstrates the promising potential of PMW MLCC for cooling application which may be considered more favorable than PST-PMW not mentioning several other advantages inherent to PMW. In this work (R. Yin et al., Adv. Funct. Mater. 2502550 (2025)), the authors further show that the heat flux of prototype cooler based on with 400 PMW MLCCs is 245 W cm⁻² which correspond to three orders of magnitude higher than the state-of-the-art current EC coolers based on PST MLCCs (Y. Wang et al Science 2020, 370, 129). Meanwhile, the voltage used to drive such high cooling performance is about 350 V corresponding to nearly half of that used in PST-PMW counterparts. This again supports the referee argument that enhanced EC strength of EC materials is crucial to cooling applications.

Regarding the practical cooling applications with respect to magnetocaloric counterparts, the referee has already commented that "In previous Nature work (Nature 575, 468 (2019) or ref. 5), the authors have made a comparison of EC effect with magnetocaloric effect in Gd under different temperatures to demonstrate the advance of EC multilayer capacitors for solid-state cooling applications. Six years later, the results on Gd are still used for comparison. Note that magnetocaloric effect in Gd has been well studied since 1976 (G. V. Brown. J. Appl. Phys., 47,3673, (1976)). There have been many advances in magnetocaloric materials, especially during the past two decades which show better magnetocaloric response than Gd (Progress in Materials Science 93, 2018, 112-232)." Now the authors add further evidence to support that Gd remains the best material for developing magnetocaloric prototypes. This may be true for the magnetic field around 1 T (Adv. Energy Mater. 2019, 9, 1901322). For the magnetic field exceeding 2 T, there are various magnetocaloric materials exhibiting higher caloric response than PST-PMW reported in current work (Adv. Energy Mater. 2019, 9, 1901322). One can comment on the costly large magnetic field of 2 T while the other can also comment on the high voltage of 600 V used here exceeding the safety voltage in laptop, cellphone and refrigerator not mentioning safety concern over wearable devices. The use of Pt electrodes is also costly. PST-PMW MLCC here is not a prototype which is the key working body. The high EC performance is pushed by high electric field close to breakdown field (750 V as claimed by authors). The EC-induced temperature drops below 1 K for the voltage below 100 V, which is much smaller than benchmark magnetocaloric Gd under 1 T. Therefore, to avoid potential overselling of electrocaloric cooling over magnetocaloric counterpart, it is reasonable to put the state-of-the-art results obtained by magnetocaloric materials under 2 T (Adv. Energy Mater. 2019, 9, 1901322). This is fair to make rational comparisons between different caloric materials given that 2 T is accessible to labs and industries. Consequently, striking advances for practical applications are not revealed in current work compared with the state-of-the-art results in PMW MLCC which exhibits numerous advantages than PST-PMW solutions.

Inconsistencies between the measured EC responses and the theoretical understandings

The mechanism for EC response in PST-PMW is also questionable. The DSC data is not consistent with dielectric results; the structural origin of increased disorder remains out of reach and the correlation between local structure and first order phase transition remains elusive. These concerns are critically linked to EC properties which are highly relevant to main claim of this work, which has not been fully addressed or just ignored by authors.

Both referee 2 and referee 3 raised concerns on the relaxor behavior. However, the authors insisted on negligible role of relaxor in EC response based on dielectric spectra under different frequencies and good agreement between direct and indirect results. The referee is not fully convinced by authors' response, which is explained as follows:

First, it is early to conclude that relaxor behavior is weak. The referee 2 has already commented that "only three frequencies were shown in Fig. S14, which are not enough to analyze the relaxor behavior" while the authors argued that "The three frequencies differ enough to show that relaxor behaviour is weak and may be ignored.". The strength of relaxor behavior can be analyzed by the shift in dielectric peak temperature ΔT_m between low and high frequencies (i.e. 100 Hz c 1 MHz in Nature Materials 17, 718–724 (2018)). Although ΔT_m between 100 Hz and 10 kHz is small, the shift between 100 Hz and 1 MHz can be big.

Based on dielectric data with limited frequency range (Fig. S14), one can see: $T_m=252.2$ K at 100 Hz and $T_m= 254.9$ K at 10 kHz for 85PST-15PMW; $T_m=258.4$ K at 100 Hz, $T_m=259.5$ K at 1 kHz and $T_m= 261.4$ K at 10 kHz for 90PST-10PMW. ΔT_m between 100 Hz and 10 kHz is 2.7 K for 85PST-15PMW which is slightly smaller than that (3 K) for 90PST-10PMW. This result indicates that relaxor behavior is weakened with less disorder with increasing PMW content which contradicts with the claim that "The small increase in thermal hysteresis on increasing PMW content is due to increased disorder." The referee is also confused that the authors have already mentioned that "The weak relaxor behaviour arises from the weak disorder in our weak solid solutions". Given that the weak relaxor behavior has been ignored, why is the increase in thermal hysteresis on increasing PMW attributed to increased disorder?

More importantly, the dielectric peak temperature- a signature of the phase transition temperature also contradicts with the results obtained by heat capacity measurement (Fig. S5). For instance, the dielectric peak temperature for 90PST-10PMW is around 260 K (Fig. S14) which is about 20 K higher than that obtained by heat capacity measurement (Fig. S5). Meanwhile, the dielectric peak temperature for 85PST-15PMW is around 252 K (Fig. S14), which is about 10 K higher than that obtained by heat capacity measurement (Fig. S5). Such large discrepancies in the phase transition temperature cannot be attributed to the difference arising from different experimental techniques.

Abnormal behavior also occurs in magnitude of dielectric constant under different frequencies. The dielectric constant at 100 Hz and 1 kHz for 90PST-10PMW is higher than those for 85PST-15PMW. By contrast, the dielectric constant at 10 kHz for

90PST-10PMW is abnormally smaller than that in 85PST-15PMW.

“The small increase in thermal hysteresis on increasing PMW content is due to increased disorder.” This explanation is also questionable. The increase in thermal hysteresis is usually indicative of strengthening of the phase transition or sharper phase transition. For instance, first order phase transition is accompanied by notable thermal hysteresis while second order phase transition exhibits weak or vanishing thermal hysteresis. The stronger phase transition implied from the increased thermal hysteresis usually results in stronger EC response in PST-PMW which contradicts with the main experimental results. The sharper phase transition caused by including PMW should give rise to higher dielectric constant with increasing PMW content, which also contradicts with dielectric spectra data (Fig. S14). The referee also refers the zero-field entropy change for the transition of $|\Delta S_0| = 13.4 \text{ kJ K}^{-1} \text{ m}^{-3}$ (85PST-15PMW) which is smaller than $|\Delta S_0| = 14.8 \text{ kJ K}^{-1} \text{ m}^{-3}$ (90PST-10PMW) in Fig. S5.

The authors simply ignored the requests on characterization of the microstructure from referee 2 (as well as referee 3). The poor quality of TEM (i.e. Fig. 1d) has been questioned according to referee 2 previous comment. Instead, TEM results with much better images are often reported in ferroelectric communities. For comparison, please refer to the EC results in modified PST (Fig. 2 in R. Yin et al., J. Adv. Ceram. 14, 9221088 (2025)) and PMW (Fig. 1 in R. Yin et al., Adv. Funct. Mater. 2502550 (2025)) whereas the quality of TEM is superior to that reported in current work. Now HRTEM is accessible to most labs in the world. TEM with high quality is highly relevant to this work as it inspires confidence in current analysis and quality of current work not to mention that the origin of order/disorder and microstructure related to increased thermal hysteresis due to including PMW remain unclear.

The authors labour the point that “weak relaxor behaviour (Supplementary Note 12) may be ignored” and “a weak relaxor behaviour notwithstanding” as “Indirectly measured values of $|Q|$ (solid lines, Fig. 3b) match well with the directly measured values” and “Indirectly measured values of $|Q|$ (solid lines, Fig. 3b) match well with the directly measured values”. The authors used Q rather than the temperature change. Note that the direct method based on thermocouple measures directly the temperature instead of the Q . The conversion of Q to the temperature requires the heat capacity that is electric field dependent. However, the field-dependent heat capacity has been ignored by authors while its role in analyzing the consistency between indirect and direct methods was not discussed in detail.

Moreover, the referee found that the claimed good agreement between indirect and direct method is also conditional. For instance, “A least squares fit between the values of $|\Delta T_j(T_s)|$ that we measured when removing $E = 8.3 \text{ V } \mu\text{m}^{-1}$, and the corresponding values of $|\Delta T(T_s)|$ (solid lines, Fig. S10d, right axis) that we identified for the active volume using the indirect method (Fig. S4c), implies $|\Delta T_j| \sim f |\Delta T|$ with $f = 0.76$. Here, f is the product of a factor f_1 for initial layer thermalization in the active area, and a factor f_2 for subsequent thermalization of the active area with the thermocouple and the affixing drop of black paint.” Following Fig. 3b-3d, the readers might believe that good agreement between indirect and direct methods is achieved without further reading how the fitting technique was imposed. These results are obtained by good fitting with a calibration factor f . The referee may treat that they are not directly consistent with each other. In addition, the measurement frequency for PE loops is not found. The frequency affects PE loops and therefore indirect measurement results. For instance, PE loops measured at high frequency may cover the loss. The PE loops at 230 K are not complete, indicating the presence of leakage. Therefore, the claim that “Indirectly measured values of $|Q|$ (solid lines, Fig. 3b) match well with the directly measured values” and “Indirectly measured values of $|Q|$ (solid lines, Fig. 3b) match well with the directly measured values” cannot be used to explicitly rule out the presence of relaxor behavior as the indirect method itself raises concerns to be technically verified. It is suggested to measure the EC response in bulk PST-PMW ceramics via both direct and indirect method (including field-dependent heat capacity) to rule out the inactive volume.

Referee #3

(Remarks to the Author)

Overall, I am satisfied with the response of Guo et al. to my queries and the changes they have implemented as a result. I have read through the revised manuscript and their additional clarifications and explanations are of benefit to the reader for understanding the main results and their potential impact for the electrocalorics field. I think their manuscript is now suitable for publication in its present form.

Referee #4

(Remarks to the Author)

I co-reviewed this manuscript with one of the reviewers who provided the listed reports.

Referee #5

(Remarks to the Author)

I co-reviewed this manuscript with one of the reviewers who provided the listed reports.

Version 2:

Reviewer comments:

Referee #1

(Remarks to the Author)

I have carefully read the exchange between Reviewer 2 and the authors in the previous round of review. Overall, I find the

authors' replies reasonable and logically consistent, and I tend to support their main arguments.

In my view, Reviewer 2 often makes comparisons that do not match the authors' central claim. The authors' key contribution is to demonstrate a macroscopic EC working body that can, in principle, cool a load through room temperature in a thermodynamically consistent way. However, many of the reviewer's critiques compare systems that cannot cool through room temperature with a work that is designed to do so, compare thin films or single-layer samples with multilayer capacitors that have enough thermal mass to act as real working bodies, focus only on EC strength at the same electric field while ignoring that real cooling depends on how large a field can actually be used in a device, over-emphasize the comparison of COP without making sure they correspond to the same temperature span, and use performance data from conditions that are not used in real prototypes to judge practical value. Because of these issues, many of the reviewer's conclusions seem to rely on mismatched criteria and do not directly test the authors' main claim.

Overall, my impression is that Reviewer 2's comments sometimes read more like a debate between different technical routes, for example favoring PMW-based systems, rather than a direct evaluation of whether the authors' approach succeeds in its stated goal. Following I list some specific examples from the exchange that illustrate these points.

Referee #3

(Remarks to the Author)

I have read the revised manuscript and authors' point-by-point rebuttal to reviewer #2 and #5, particularly regarding novelty, and believe that they have been answered to a satisfactory degree that the article can now be published in Nature.

The referee argues that the work does not represent a significant advance, claiming that existing material systems (specifically modified PST ceramics and PMW-based MLCCs) already demonstrate comparable or better electrocaloric performance under similar electric fields, that the mechanism of using a supercritical phase transition and tuning the phase transition temperature via solid solution is not new, and that the proposed PST-PMW composition does not offer a practical advantage over state-of-the-art PMW MLCCs. The reviewer also challenges the key claim of Guo et al. being the first to demonstrate an EC body capable of cooling through room temperature, citing R. Yin et al.'s work as a prior example in relevant temperature ranges.

The authors clarify their breakthrough and defend the novelty by emphasizing that their core advance is the first macroscopic, unannealed electrocaloric material system that operates above and below room temperature, thereby enabling cooling through it. The author's point out that Yin et al.'s modified PST samples are unsuitable for prototypes due to the low thermal mass associated with a single layer geometry (rather than MLCC), despite the fact they may potentially cool through room temperature. In response to the reviewer's claims that the same behaviour is potentially achievable in PMW in the desired MLCC geometry, the authors give good arguments that PMW is in fact thermodynamically incapable of cooling through room temperature because its electrocaloric effect changes sign, as discussed in Supplementary Note 1 with exemplar thermodynamic cycles to prove this point. This is an important argument that maintains the high novelty of Guo et al.'s work, even when the work of R. Yin et al. is taken into account [minor point: the step 4 to 1 text in Supplementary Note 1 should presumably read "PMW is translated to the cold end with no field applied."] The authors also clarify that their materials design strategy of using PMW to disrupt dipolar order (hence lowering transition temperature) without destroying the B-site order (therefore preserving the latent heat) is novel for this unexplored solid solution and avoids the need for long anneals.

The referee raises concerns about performance of the PST-PMW MLCCs compared to state of the art PMW MLCCs, namely that the latter has higher efficiency (COP), higher heat flux, and lower operating voltage. (i.e. higher electrocaloric "strength"). Regarding the COP, the authors revisit and correct their own data visualisations and note that at 300 V their materials' COP actually exceeds the value cited for PMW. For the heat flux performance, they make the reasonable point that comparing the heat flux of a single PST-PMW to a stack of 400 PMW MLCCs seems unfair since it is not a like-for-like comparison. On the final point, the authors also make a reasonable rebuttal that comparing electrocaloric strength at the same field is secondary to the key parameter for cooling which is the absolute size of the electrocaloric effect (ΔT) using the highest possible field, enabled by the MLC design.

The reviewer #2 raises several perceived technical concerns which are also addressed in the revised manuscript and rebuttal document. Discrepancies between phase transition temperatures identified via dielectric spectroscopy and heat capacity measurements were acknowledged and addressed by the authors through new, less noisy dielectric spectra on both cooling and heating and explained in terms of a known coexistence of first-order and relaxor behaviour. The authors have revisited their claims of "weak" relaxor behaviour and present new data showing the frequency shift of the dielectric peak and link increased PMW content to increased disorder, stronger relaxor character, and larger thermal hysteresis. The reviewer #2 also challenges the quality and extent of microstructural investigation by transmission electron microscopy, and while these criticisms have validity, the authors are also reasonable in stating that it is not required to support their central claim. The need for a calibration factor to ensure consistency between direct and indirect measurements is queried by the reviewer but this is well defended by the authors, as the role of inactive material has to be accounted for in direct measurements (but is not relevant for the indirect electrical measurements). They also clarify that the field-dependence of the heat capacity is not ignored in their method and is encompassed within their entropy mapping approach.

Overall, the authors have provided substantive responses to the criticisms and made concrete revisions to the manuscript

and supplementary information. I believe that they have done a good job responding to the skepticism regarding the novelty and proposed practical superiority of PST-PMW solid solution over PMW.

Referee #4

(Remarks to the Author)

I co-reviewed this manuscript with one of the reviewers who provided the listed reports.

Referee #1

We thank the referee for the detailed feedback, which we have used to improve the manuscript.

All changes (except minor figure changes) are shown using **yellow** highlights, or by highlighting a word in **blue** to indicate that all subsequent text in the paragraph is substantially or wholly new - this avoids the need to read large chunks of highlighted text.

These changes include changes due to feedback from all referees, changes where we found ways to improve the presentation, and changes that we made to comply with the Editorial advice to shorten the paper. Note that the improved presentation has led us to swap the order of Supplementary Notes 18-20.

Minor figure changes are as follows:

- In Fig. 4, using the feedback of referee 2, we have extended PST data below the Curie temperature using indirect data in order to show that PST performs well only above room temperature. These indirect data are shaded pale, and so the other data are no longer presented with (avoidable) variable shading (we have likewise removed variable shading from Fig. S17). We also used the feedback of referee 2 to repeat Fig. 4 in new Supplementary Note 15 and add data for other state-of-the-art magnetocaloric materials that are being used in prototypes.
- In Fig. 5f, we have improved the way we present the shading and made a proper scale bar. Equivalent changes have been made in what is now Fig. S29 (whose panels have been swapped).

Also, at the request of the Editor, the title has been shortened, and the two Tables have been moved to Extended Data.

The authors present a promising MLC based on solid solutions of PST and antiferroelectric PMW, which demonstrates large EC effects above and well below room temperature. Compared with PST MLCs prototypes, the PST-PMW MLCs reported in this work show impressively low Curie temperature (~230 K), addressing application challenges of EC effects in low temperatures. Overall, this study presents excellent research findings and can be accepted with a few revisions to the manuscript and the Supporting Information.

We thank the referee for these positive comments.

Major Comments:

1- To help readers better evaluate the practical cooling capability of the reported MLCs, I recommend including the thermal conductivity of the PST-PMW, either as measured or from reasonable estimates. Additionally, a brief calculation converting the measured temperature change ($\Delta T \sim 3$ K) into an estimated heat flux at a typical EC system operation frequency would provide meaningful context. While the EC temperature shift reflects intrinsic material performance, the actual heat flux also

depends on how quickly heat can be absorbed or dissipated from the active region. Including this analysis would add value from a device engineering standpoint.

In the revised text below, we now convert the heat flux pumped by our cooling cycles (Q in Fig. 5c) into an estimated cooling power, which we explain is not limited by ceramic thermal conductivity (underlined text represents additional new information).

“The active-volume normalized heat $Q = \int_4^Y TdS'$ (Fig. 5c) pumped from a load at the cold end of the regenerator (blue area, inset of Fig. 5b) varies with cold temperature T_c , and also temperature span $T_h - T_c$ given the balanced regeneration. The maximum heat ($|Q| = 9.3 \text{ MJ m}^{-3}$) necessarily falls just short of the maximum isothermal heat ($|Q| \sim 10 \text{ MJ m}^{-3}$, Fig. 3b), and corresponds to a maximum cooling power of 77.7 mW MLC⁻¹ given our active volume (32.6 mm³, Methods) and a typical operational frequency (0.25 Hz) that is limited by dielectric fluid rather than ceramic thermal conductivity³.”

Ceramic thermal conductivity is therefore a moot point, but we can say that a similar ceramic is known to have a thermal conductivity of $\sim 1 \text{ W m}^{-1} \text{ K}^{-1}$ (<https://doi.org/10.1063/1.125483>).

2- While the authors describe a direct measurement of EC temperature change using a thermocouple affixed to the MLC surface, several key experimental details remain unclear.

We wrote that “Direct measurements of EC temperature change... were performed using a thermocouple affixed with a small drop of black paint to the MLC face centre (Methods)”, and we humbly wonder if the Referee might have not checked our (now-improved) text in Methods?

Given the importance of this measurement in validating the EC performance, it would strengthen the manuscript to provide additional information on

the thermocouple type,

We wrote in Methods that the thermocouple is K-type, and we have now added details of the manufacturer and geometry, such that we now refer to “a K-type thermocouple (THERMA Thermofuehler GmbH) with junction diameter 400 μm and wire diameter 80 μm .”

response time,

We wrote in Methods that the thermocouple was “monitored at $\sim 4.9 \text{ Hz}$ using a Keithley 2110 Multimeter”.

the thermal contact quality between the thermocouple and the sample (e.g., thermal interface resistance introduced by the black paint? The black paint thermal conductivity?),

There is nominally no black paint between thermocouple and sample as the thermocouple presses on the sample before the drop of black paint is added to keep it in place. To help remind the reader, and distinguish from the scenario in which a layer of black paint is added for IR camera measurements, we have changed:

“thermalization of the active area with the black paint and the thermocouple”

to

“thermalization of the active area with the thermocouple and the affixing drop of black paint”.

Therefore heat flow between thermocouple and sample may instead depend primarily on the interfacial resistance between thermocouple and sample, but it may nevertheless be influenced by black paint thermal conductivity and associated interfacial resistance.

The parameters discussed above are unknown, and would be useful for thermal modelling, but any such modelling would inflate the values of the temperature jumps that we measure, and it would not be clear whether we had inflated them beyond the correct values. So it is best to be conservative and report measured temperature jumps, as we do.

and the thermal environment during measurement.

Please see the section in Methods called “Set-up for highly adiabatic measurements at different starting temperatures”, now heavily revised for clarity, completeness, accuracy, and continuity with respect to the previous Methods section on which it builds, namely “Highly isothermal measurements of heat”. This previous section has been much improved.

For instance, was the sample mounted under thermal insulation to minimize heat loss?

There is no thermal insulation.

Additionally, while the measurements were conducted down to 240 K, the manuscript does not specify how the low-temperature environment was achieved or stabilized.

Temperature control is described in the Methods section called “Highly isothermal measurements of heat”, and the description therein is relevant for the subsequent Methods section called “Set-up for highly adiabatic measurements at different starting temperatures”.

Clarifying the cooling method and temperature control accuracy would enhance the credibility and reproducibility of the results.

We hope the Referee finds our (revised) presentation in Methods acceptable.

The temperature changes that we report are consistent with our indirect measurements (Fig. 3d), and these indirect measurements are consistent with our isothermal heat measurements (Fig. 3b).

Moreover, we have previously shown that thermocouple measurements from our team match well with infra-red measurements (ref. 5). To explain why we do not have infrared imaging here, we stated that “infrared imaging is challenging below room temperature”. Note that infrared imaging would give slightly larger temperature changes than thermocouple measurements because, as explained in ref. 5, there is less inactive thermal mass and the measurement is faster.

Minor suggestion

1. **"CrossleyPRX" appears on page 5.**

Corrected to cite ref. 39, which had already been cited, so no need to reorder references.

2. Check the consistency of the formatting for T_c on page 8.

Following standard practice, we use T_C to denote Curie temperature, and we use T_c to denote the cold temperature in cooling cycles. Having defined T_C at the start of the paper, we now clarify the distinction between the two symbols when we define T_c by adding:

“(T_c differs from Curie temperature T_C)”.

3. Include legends for the curves in Figs. 3 c-d and S10 c-d to enhance readability.

For each of these figures, the legends for panels c-d are the same as the legend that we give for panel b. We believe that it is best to avoid repeating the legend from panel b in panels c-d, partly so the reader can use the common legend without needing to check for matching values for themselves, and partly because there is no space in panel d.

4. The language could be further refined by avoiding subjective statements like "We see" and "Let us now compare" in the discussion.

We have removed all of the cited statements as follows:

First:

“We see from Fig. 3a that”

... is now:

“Fig. 3a shows that”.

... and likewise in Supplementary Information with Fig. S10a.

Second:

“We see that”

... is now more informatively:

“Even somewhat away from the upper thermalization limit, MLCs of both PST-PMW compositions show”

Third:

“Let us now compare direct and indirect measurements of temperature change.”

... has been deleted in the main paper and in Supplementary Note 8, as it is now superfluous given that the subsequent sentence now explicitly uses the words “direct” and “indirect” to describe the measurements of temperature change that we compare.

Fourth: “We see that” has been eliminated in our rewrite of Supplementary Note 18 (formerly Supplementary Note 19).

5. The title of the SI is different from that of the main text.

The title of the SI now matches the revised title of the main text, which was shortened as required by the Editor.

Referee #2

We thank the referee for the detailed feedback, which we have used to improve the manuscript.

All changes (except minor figure changes) are shown using **yellow** highlights, or by highlighting a word in **blue** to indicate that all subsequent text in the paragraph is substantially or wholly new - this avoids the need to read large chunks of highlighted text.

These changes include changes due to feedback from all referees, changes where we found ways to improve the presentation, and changes that we made to comply with the Editorial advice to shorten the paper. Note that the improved presentation has led us to swap the order of Supplementary Notes 18-20.

Minor figure changes are as follows:

- In Fig. 4, using the feedback from the referee (referee 2), we have extended PST data below the Curie temperature using indirect data in order to show that PST performs well only above room temperature. These indirect data are shaded pale, and so the other data are no longer presented with (avoidable) variable shading (we have likewise removed variable shading from Fig. S17). We also used the feedback of referee 2 to repeat Fig. 4 in new Supplementary Note 15 and add data for other state-of-the-art magnetocaloric materials that are being used in prototypes.
- In Fig. 5f, we have improved the way we present the shading and made a proper scale bar. Equivalent changes have been made in what is now Fig. S29 (whose panels have been swapped).

Also, at the request of the Editor, the title has been shortened, and the two Tables have been moved to Extended Data.

EC effects in ferroelectrics have attracted growing interest for cooling applications. The authors studied EC effect in PST-PMW multilayer capacitors which shows the existence of large EC responses above and below room temperature. Based on their previous works using PST (Nature 575, 468 (2019) or ref. 5), they introduce PMW to form solid solutions. They show that the phase transition temperature is markedly lower from 290 K to 230 K. As a result, large EC effects occur at lower temperatures, expanding the operating temperature window which has not been accessible in their previous Nature work (Nature 575, 468 (2019) or ref. 5) as their data were limited to above room temperature. Meanwhile, the efficiency of PST-PMW multilayer capacitors remains slightly larger than PST. With a comparable EC response and efficiency, PST-PMW requires no long-term annealing like PST and exhibits a wider working temperature window which offers a promising alternative to replace PST used as the key elements for design of EC prototype devices. Overall, the work on PST-PMW represents advances over PST according to EC data measured at different temperatures and electric fields through both direct and indirect methods.

We thank the referee for these positive comments.

Despite these improvements to the previous Nature work published in 2019, the results obtained in the current work fail to demonstrate significant advances compared with the state-of-the-art studies including modified PST ceramics, lead-free and lead-contained EC multilayer capacitors, and previous results on multilayer capacitors based on PST solutions.

Our main claim is the demonstration of an EC working body that could be used to cool a load through room temperature, and our revised title “Electrocaloric effects across room temperature in multilayer capacitors” now makes that more clear.

In our original manuscript, we primarily made comparison with the 2019 *Nature* work (ref. 5) because it reported MLCs of PST that were subsequently used by several groups to make the state-of-the-art prototypes (that therefore operate wholly above room temperature only, refs 1-4).

In the revised manuscript, paragraph 2 of the main text has been rewritten to explain why samples such as those referred to by the referee cannot be used for cooling to below room temperature:

“EC effects in existing material systems cannot be exploited for cooling through room temperature. For PST modified via B-site disorder³² or solid solution³³, sub-room-temperature EC effects were reported in thin single layers (~100 μm) and are small (≤ 1.6 K). For other ceramics, sub-room-temperature EC effects are sometimes larger ($\leq 1-4$ K), but they arise in thin single layers³⁴, are compromised by substantial inactive thermal mass³⁵, or decay to zero and change sign on crossing room temperature (as seen for antiferroelectric PMW)³⁵⁻³⁷. (Ref. ³⁴ also demonstrated an MLC that is compromised by substantial inactive thermal mass, and was only measured at room temperature.)”

³² R. Yin *et al.*, *J. Adv. Ceram.* **14**, 9221088 (2025).

³³ L. Shebanov and K. Borman, *Ferroelectrics* **127**, 143 (1992).

³⁴ F. Du *et al.*, *Nature* **640**, 924 (2025).

³⁵ R. Yin *et al.*, *Adv. Funct. Mater.* 2502550 (2025).

³⁶ S. Hirose *et al.*, *J. Phys. Energy* **5**, 035009 (2023).

³⁷ J. Li *et al.*, *Adv. Funct. Mater.* **31**, 2101176 (2021).

In the above paragraph that we quote, two new references are new [32,34]*, two previously cited references are deleted**, and some of the references have been reordered in light of our revisions to the text.

* Ref. 32 was published after our original submission.

** The two deleted references are:

- L. Shebanov and K. Borman, On lead-scandium tantalate solid solutions with high electrocaloric effect. *Ferroelectrics* **127**, 143 (1992). This is replaced with new ref. 32, which reports larger EC effects in PST with B-site disorder.

- L. A. Shebanov, E. H. Birks, and K. J. Borman, X-ray studies of electrocaloric lead-scandium tantalate ordered solid solutions. *Ferroelectrics* **90**, 165 (1989).
... from which no specific results were cited.

Moreover, the manuscript only focused on comparison between PST-PMW and PST (Nature 575, 468 (2019) or ref. 5) by still keeping the data on PST already published without expanding them to lower temperature at least for better comparison. As one can see from Fig. 4, one may expect a better EC efficiency and possibly a wider working temperature window for PST alone.

As explained above, we primarily made comparison with the 2019 *Nature* work (ref. 5) because it reported MLCs of PST that were subsequently used to make the state-of-the-art prototypes, which operate wholly above room temperature (refs 1-4).

Fig. 4 only showed PST data down to the room-temperature Curie temperature because it is based on directly measured EC effects that are very small below this temperature, such that they were not measured in the experiments associated with ref. 5.

However, we see from the feedback that it is important to show in Fig. 4 that EC effects in PST become small below room temperature. We therefore now show this by using indirectly measured PST data from ref. 5 to extend the PST data in Fig. 4 below room temperature (pale purple in the revised Fig. 4 below).

Besides these PST-based systems, the state-of-the-art results are not considered. Indeed, there have been several recent publications reporting the EC cooling near and below room temperature which is comparable or larger than the results obtained in this manuscript. Therefore, the main novelty claiming large electrocaloric effects above and below room temperature is significantly diluted.

This matter is addressed in our response to the earlier comment that begins “Despite these improvements...”.

Note that the same group has already published EC responses, while not as high, below room temperature in another system (Nature Materials 23, 639 (2024))

The 2024 work by a partially overlapping set of authors describes EC effects in an ultra-thin (60 nm) epitaxial film. These EC effects are enhanced by 1-2 orders of magnitude due to strain from the substrate. The ultra-thin film on a substrate cannot be used for applications. The work is therefore not relevant to our paper.

and there are already published works showing high EC properties below room temperature. The consideration of the existing literature, in addition to PST-based systems, must be also considered.

This matter is addressed in our response to the earlier comment that begins “Despite these improvements...”.

Moreover, this work lacks sufficient fundamental insights into the phase diagram of PST-PMW including composition, temperature and electric field with emergence of relaxor behavior which is essential to design large EC response.

We present field-temperature phase diagrams for two compositions of PST-PMW (Fig. 2 for 85PST-15PMW, Fig. S4 for 90PST-10PMW), while the equivalent phase diagrams for pure PST appear in ref. 5. Explicit presentation of a composition axis is not relevant for our demonstration of EC working bodies that could be used in a prototype to cool a load through room temperature.

It is incorrect to say that “relaxor behavior... is essential [for a] large EC response”. As shown in ref. 5 via both experimental data and Landau theory, a large EC response can be achieved (over a wide range of temperatures) without relaxor behaviour by driving a first-order transition supercritically. Likewise, as we stated in our original manuscript, we “supercritically drive a first order ferroelectric phase transition above a low Curie temperature”. The statement in our revised manuscript includes more information, and reads:

“The operating temperature range is wide because a large electric field supercritically drives the first-order ferroelectric phase transition above a Curie temperature that is suppressed due to PMW disruption of PST dipolar order.”

In particular, the local structures related to relaxor phase caused by PMW and the correlations with EC response remain unknown.

We explained in the main paper that:

“Indirectly measured values of $|Q|$ (solid lines, Fig. 3b) match well with the directly measured values, a weak relaxor behaviour notwithstanding (Supplementary Note 12)”.

However, the comment (and a related comment from Referee 3) reveals that our text did not make it clear that match means the weak relaxor behaviour may thus be ignored.

We have therefore changed the text quoted above to read:

“Indirectly measured values of $|Q|$ (solid lines, Fig. 3b) match well with the directly measured values, such that weak relaxor behaviour (Supplementary Note 12) may be ignored.”

The same change has been made in Supplementary Note 8.

We also now explain why the relaxor behaviour is weak by writing in Supplementary Note 12 that “The weak relaxor behaviour arises from the weak disorder in our weak solid solutions”, and we add directly afterwards that it “may be ignored” so that this useful information is also present in that Supplementary Note.

In addition, other technical issues such as the conditions for calculating efficiency

The efficiency calculation was explained in the main text section on “EC cooling cycles”. Without further detail, we cannot understand what the referee finds unsatisfactory.

and the comparison between PST-PMW and Gd rather than state-of-the-art magnetocaloric materials remain yet to be addressed.

It is a good idea to include data for state-of-the-art magnetocaloric materials, even though they do not outperform Gd. Therefore, having presented the performance of Gd in prototypes (red line in Fig. 4), we now explain that “Magnetocaloric working bodies based on other materials show similar/inferior performance (Supplementary Note 15)”. The new figure below appears in new Supplementary Note 15, which is entitled “Comparison of MLCs with magnetocaloric working bodies”.

Therefore, based on these considerations, the reviewer cannot recommend the publication of the current manuscript in Nature.

We request that the reviewer might please reconsider this recommendation in light of our responses.

In the following, please find some comments and concerns the authors may consider if they wish to submit their work somewhere else.

#1 The authors wrote “Consequently, a growing number of prototype EC coolers that exploit assemblies of thin ceramic/polymer layers (MLCs)1-4,15-24 and polymer bilayers25-29/monolayers30,31 can cool loads down to temperatures that lie near room temperature. However, EC cooling through room temperature is yet to be achieved, even though such cooling is important for food, beverages, the built environment, and medicine, etc.”

Here the central comment on previous studies raised by the authors is that previous works mainly focused on the working temperature range near or above room temperature.

Please note that the word “mainly” should read “exclusively” because no previous EC prototype cooled a load below room temperature.

Moreover, the main novelty based on this argument is that they have achieved EC cooling through room temperature.

We present the first working body in which large EC effects can be driven over a wide range of temperatures that include room temperature. Therefore the working body we present can be used in prototypes for cooling through room temperature, and likely will be, mirroring the development of prototypes that followed the 2019 report on MLCs of PST.

This argument is not fully true as a few recent works reported large EC properties through room temperature in various EC materials. For instance, a very recent work (J Adv Ceram 2025, 14: 9221088. <https://doi.org/10.26599/JAC.2025.9221088>) entitled “Optimizing electrocaloric effect of PbSc_{0.5}Ta_{0.5}O₃ ceramics near/below room temperature by ordering degree modulation” reported that through tuning of ordering degree the working temperature range is lowered near and below room temperature. It reported an EC-induced temperature change of 1.5 K under an electric field of 6.0 V/μm at -10°C (or 263 K) for $\omega = 0.51$ which is comparable that obtained in PST-PMW under the same temperature and field. Meanwhile, this recent work used lattice ordering to tune the working temperature range for achieving large EC effect in PST ceramics from 40°C (or 313 K) to -10°C (263 K).

As discussed in our response to the earlier comment that begins “Despite these improvements...”, we now cite this paper as ref. 32, and explain that the sub-room-temperature EC effects it reports cannot be exploited for cooling through room temperature because they arise “in thin single layers (~100 μm) and [are] small (≤1.6 K)”.

Besides, the fabrication of modified PST ceramics does not require long-term annealing as required in previous Nature work (Ref. 5).

Previously demonstrated “modified PST ceramics” do not require a long anneal but they were not used in the state-of-the-art prototypes (refs 1-4). Our work suggests that it would be attractive to replace the MLCs of PST in state-of-the-art prototypes with MLCs of PST-PMW, not just to achieve cooling through room temperature, but also to avoid the

time-consuming and expensive anneal. Both of these issues are important in the push for commercial applications.

Therefore, the current manuscript shows the absence of striking advances over this recently published work as they both show the same level EC response, and both involve the operating temperature below room temperature. For instance, for modified PST, ordering degree was used to shift phase transition temperature towards lower temperatures through room temperature while for PST-PMW, incorporation of PMW lowers the phase transition below room temperature. This strongly challenges the argument by the authors that “EC cooling through room temperature is yet to be achieved” and “an energetically expensive 42-day anneal is required to achieve good B-site order”.

We request that the reviewer might please reconsider this analysis in light of the above response to comment #1.

#2 The authors wrote “For example, indirect measurements show that a thin plate of B-site disordered PST exhibits an adiabatic temperature change of magnitude $|\Delta T| = 0.6$ K at 260 K, while a thin plate of PST-PbSc_{0.5}Sb_{0.5}O₃ exhibits $|\Delta T| = 1.5$ K at 276 K.”

In this sentence, the authors recognized that the phase transition temperature of PST has been lowered below room temperature through solid solution method (Ferroelectrics 127, 143 (1992) or ref. 32).

We are not claiming that the solid-solution method is novel, but please note that solid solutions of PST and PMW are “hitherto unexplored”.

Moreover, the authors overlooked the electric field used to trigger the large EC-induced temperature change of 1.5 K at 276 K. It is found that the electric field used in this previous work (Ferroelectrics 127, 143 (1992) or ref. 32) was 25 kV/cm as identical as 2.5 V/ μ m. This field is much lower than that used in current work. For instance, under the same electric field of 2.5 V/ μ m, the temperature change is about 0.8 K which is less than half of that reported in previous work (Ferroelectrics 127, 143 (1992) or ref. 32).

For the MLCs of 85PST-15PMW on which we primarily focus in the main paper, our indirect EC measurements show that a field change of 2.5 V μ m⁻¹ yields a peak of $|\Delta T| \sim 1.8$ K (Fig. 2c), not 0.8 K as stated by the referee. Correcting 1.8 K for layer thermalization using the factor of 0.85 that we give in the main paper implies an effective temperature change of $|\Delta T_{\text{eff}}| \sim 1.5$ K, which is similar to the value quoted by the referee.

However, as we have explained earlier, the value quoted by the referee was obtained for a thin plate that “cannot be exploited for cooling through room temperature”, and so the comparison is moot.

Note that we discussed the magnitude of the electric field by writing that we use “the same voltage (600 V) that is used to repeatedly drive the geometrically similar MLCs of PST in prototypes¹⁻⁴”. (In the revised manuscript, we have deleted the “the” before “geometrically”.)

Meanwhile, the operating temperature for PST-PbSc_{0.5}Sb_{0.5}O₃ is from -20°C to 10°C which also covers the temperature through room temperature, in contrast with the

claim that “However, EC cooling through room temperature is yet to be achieved”. Both studies use the solid solution method to mainly introduce modification on B-site: the previous one used PST-PbSc_{0.5}Sb_{0.5}O₃ while this work uses PST-PMW.

This matter was addressed earlier in two of our responses that we reproduce below:

1. We present the first working body in which large EC effects can be driven over a wide range of temperatures that include room temperature. Therefore the working body we present can be used in prototypes for cooling through room temperature, and likely will be, mirroring the development of prototypes that followed the 2019 report on MLCs of PST.
2. We are not claiming that the solid-solution method is novel, but please note that solid solutions of PST and PMW are “hitherto unexplored”.

The smaller EC strength in current work than previous one strongly declines the advance of current work given that EC strength of EC materials is crucial to practical cooling applications.

After correcting the referee’s estimate of 0.8 K to 1.5 K two answers ago, one finds that the two EC strengths are equivalent. However, EC strength is of secondary importance and does not affect our main claim to have demonstrated an EC working body that could be used to cool a load through room temperature.

#3 PMW is known as an antiferroelectric and EC studies in PMW were conducted by several groups. For instance, PMW including both bulk ceramics (Adv. Funct. Mater. 2021, 31, 2101176) and multilayer capacitors (Adv. Funct. Mater. 2025, 2502550 DOI:10.1002/adfm.202502550) showing large EC temperature change have been studied. To make a direct comparison, here we mainly focus on PMW multilayer capacitors under the same electric field of about 21 V/μm. This previous work shows a large temperature change of 4.0 K at 100°C (or 283 K) which remains 2.9 K at -100°C (or 263 K). This adds another example challenging the authors’ claim that “However, EC cooling through room temperature is yet to be achieved”. For comparison, in the present manuscript, PST-PMW shows a temperature change of about 3.3 K at 283 K and about 3.2 K at 263 K driven by an electric field of 21.4 V/μm. These results indicate that the EC effects obtained under the same electric field and temperatures in PST-PMW are comparable to those obtained in PMW multilayer capacitors published recently (Adv. Funct. Mater. 2025, 2502550 DOI:10.1002/adfm.202502550). Moreover, PMW at near room temperature shows a larger temperature change of about 6.7 K at 293 K which is twice as large as that of PST-PMW in the present work. Also note that as PMW is antiferroelectric, it exhibits a combination of both negative (converse) and positive (direct) EC response depending on the working temperature. In previous work (Adv. Funct. Mater. 2025, 2502550 DOI:10.1002/adfm.202502550), it is demonstrated that COP of PMW multilayer capacitors is much higher than PST counterparts. As efficiency of PST-PMW multilayer capacitors is only slightly larger than PST multilayer capacitors, COP of PST-PMW multilayer capacitors is nearly the same as that of PST multilayer capacitors. Therefore, PMW multilayer capacitors may compete more favorably than complex PST-PMW multilayer capacitors, as already it shows large EC effect through room temperature and do not need long-time annealing. This again challenges the authors’ argument in the introduction part “However, EC

cooling through room temperature is yet to be achieved, even though such cooling is important for food, beverages, the built environment, and medicine, etc.”.

Our claim that “EC cooling through room temperature is yet to be achieved” unambiguously refers to a load being cooled through room temperature. The MLCs of PST-PMW that we present pave the way towards that goal. (To save words, we have changed “yet to be achieved” to “remains elusive”.)

PMW cannot be used to achieve the goal of cooling a load through room temperature because the EC effects “decay to zero and change sign on crossing room temperature”, as we now explain in paragraph 2 of the main text.

Our revised title “Electrocaloric effects across room temperature in multilayer capacitors”, with “across” replacing “above and below”, reflects this exclusion of PMW.

Although moot, note that EC effects in PST-PMW persist to more extreme temperatures than EC effects in PMW.

#4 It is suggested to add the temperature-composition-electric field phase diagram of PST-PMW which is crucial to understanding the EC behavior in PST-PMW

A detailed phase diagram is not relevant for our main claim, but would make interesting future work.

Huge efforts were made by the authors in characterizing the EC properties.

We thank the referee for this positive comment.

However, fundamental insights allowing to inspire further material design remains lacking in the current version of manuscript.

As the referee will know, we explained our materials design strategy in the “Discussion and outlook” section, and we finished that section, and thus the paper, by suggesting that “our materials development strategy (valence-size-sintering) should inspire improvements in other ceramic materials, for EC applications and beyond.” Note that in the revised manuscript, we have modified this sentence by adding “, ferroelectric-antiferroelectric solid solution” before the closing bracket.

As mentioned in comment #3, PMW is antiferroelectric differing from ferroelectric-like PST with ordered B-site cation order. How the symmetry phase, state, and phase transition evolve with incorporation of PMW-that is crucial to EC response-remains elusive based on the current data. Temperature-dependent characterization of phase transition behavior based on XRD and dielectric spectra is required. For instance, the temperature dielectric spectra can be used in complementary to XRD or other techniques to analyze the phases and the critical temperatures.

These are interesting suggestions for future work, but they are not required to substantiate our main claim.

In this regard, only three frequencies were shown in Fig. S14, which are not enough to analyze the relaxor behavior.

The three frequencies differ enough to show that relaxor behaviour is weak and may be ignored.

As stated earlier, we now explain in Supplementary Note 12 why there is any relaxor behaviour, and why it is weak, by writing that “The weak relaxor behaviour arises from the weak disorder in our weak solid solutions.”.

In addition, it is shown in Fig. S14 that in 85PST-15PMW it seems that there is another transition, given the kink on the left side of the dielectric peak maximum. The losses could also help the authors to identify the real behavior of the dielectric response here. There is also a strange behavior (could be related to defects) related to the frequency changes of the maximum of the dielectric constant when both 90PST-10PMW and 85PST-15PMW are compared. The abnormal/strange dynamic dielectric behavior here stays unclear and require more efforts, not mentioning the effect of applied dc-electric field.

This issue is not understood, and could be interesting to pursue further, but it has no implications for our main claim.

#5 The local structure of PST-PMW with inclusion of PMW remains elusive based on current data. For instance, the authors mentioned that “Good B-site cation order in 85PST-15PMW is confirmed via HAADF-STEM”. However, HAADF-STEM data shown in Fig. 1d and Fig. S3 are of low quality compared with previous work on PMW (Adv. Funct. Mater. 2021, 31, 2101176; Adv. Funct. Mater. 2025, 2502550 DOI:10.1002/adfm.202502550). It is suggested that STEM with high quality image should be provided.

The local structure remains elusive in so far as there is disorder amongst the low-valence cations (Sc^{3+} and Mg^{2+}) and disorder amongst the high-valence cations (Ta^{5+} and W^{6+}).

Although we commented explicitly on this disorder when we wrote of HAADF-STEM data that:

“Although the B1 columns contain Sc^{3+} and Mg^{2+} ions with quite different atomic numbers, similar intensities suggest good mixing.”.

... the comment has inspired us to recognise the disorder everywhere by:

- Changing in the abstract:
“B-site order”
to:
“B-site order between high-valence and low-valence cations”
- Explaining in the introduction that our B-site order is:
“between high-valence cations (Ta^{5+} , W^{6+}) and low-valence cations (Sc^{3+} , Mg^{2+})”

- Adding “between high-valence cations and low-valence cations” after all relevant instances of “B-site order”. The concomitant changes in Supplementary Note 1 are as follows:
 - Supplementary Note 1 is now entitled “B-site order between high-valence and low-valence cations via X-ray diffraction”.
 - Supplementary Note 1 now opens by explaining that:
 - “Here, for both PST-PMW compositions, we evaluate the B-site order between high-valence cations (Ta⁵⁺, W⁶⁺) and low-valence cations (Sc³⁺, Mg²⁺)”.
 - We have changed “B-site order S_{111} ” to “ S_{111} ”.

However, the statement that local structure remains elusive is too sweeping given that our XRD and HAADF-STEM data evidence the good B-site order between high-valence and low-valence cations.

Regarding HAADF-STEM quality, our HAADF-STEM parameters (see Methods) were set to minimise beam-induced effects in the specimen, and led to images of sufficient quality for the structural analysis that we describe above.

In addition, the authors recognized the presence of local chemical disorder by saying that “B2 columns comprise the high-Z elements (Ta⁵⁺ with Z = 73, W⁶⁺ with Z = 74) and that the B1 columns comprise the low-Z elements (Sc³⁺ with Z = 21, Mg²⁺ with Z = 12)”.

We do not understand the sense of this comment, but note that HAADF-STEM averages along atomic columns.

Meanwhile, temperature-dependent dielectric spectra show the existence of relaxor behavior. How do the local chemical disorders inherent to relaxor behavior correlate with EC response? This is also critical to understanding a very broad EC response with respect to temperature.

As stated earlier, the relaxor behaviour is weak and may be ignored (Supplementary Note 12), while the large EC response across a wide range of temperatures arises because we “supercritically drive a first order ferroelectric phase transition above a low Curie temperature”. As stated earlier, the statement in our revised manuscript includes more information, and reads:

“The operating temperature range is wide because a large electric field supercritically drives the first-order ferroelectric phase transition above a Curie temperature that is suppressed due to PMW disruption of PST dipolar order.”

#6 The cycle efficiency. The use of cycle efficiency might potentially mislead the readers as the number as large as 70-90% is calculated under ideal conditions.

We specified the use of an “ideal fluid regenerator” at the end of the introduction, in the section on EC cooling cycles, and in the Fig. 5 caption. However, we failed to specify an ideal fluid regenerator in the abstract. Therefore in the abstract we have changed:

“In addition, we show that cycle efficiencies can reach 70-90%.”

to:

“Increasing temperature span with an ideal fluid regenerator would yield cycle efficiencies of 70-90%.”.

Also, the word “prototype” should be indicated specifically to readers as multilayer capacitors are the key elements used in EC cooling prototype devices. Therefore, strictly speaking, multilayer capacitors alone cannot fully represent the prototype devices. Here the reviewer’s point is that the performance of real prototype devices based on multilayer capacitors is more convincing to demonstrate the high-efficient cooling applications.

The referee has not said explicitly where we should add “prototype”, so we can only assume that this comment follows on from the previous comment about “70-90%” efficiencies and thus refers to the abstract, which is the only place where “70-90%” appears. However, the next sentence in the abstract already uses the word “prototype” by stating that “MLCs of PST-PMW should now replace MLCs of PST in EC prototypes for efficient EC cooling through ambient”. This statement makes it clear that MLCs do not “fully represent the prototype devices”.

Indeed, a lot of EC device publications claimed high COP or efficiency while they did not specially show readers the conditions how high COP or efficiency is achieved. For instance, an Erratum for a recent Science paper (<https://www.science.org/doi/10.1126/science.adz0508>) say that “Hence, the maximum coefficient of performance of the electrocaloric heat pump—which is affected both by the cooling power and the recoverable energy—reaches 54% of Carnot’s efficiency if 99% of the recoverable energy is harvested.” Like the efficiency calculated here, cooling power based on PST multilayer capacitors is calculated under the condition only if 99% of the recoverable energy is harvested. Unfortunately, readers are unaware of the real percentage of recoverable energy under the real conditions. As these numbers are calculated based on equations rather than direct measurements on real prototype devices, using incorrect equations may yield overestimation of true values, misleading the readers and the field.

More specifically, the authors assume work recovery by following the ref. 21. More information should be provided regarding work recovery. What is the real value of the work recovery in real devices? Please provide detailed information on how to technically testify the work recovery and its role in affecting efficiency to avoid potential overestimation of the device performance.

Our COP and efficiency calculations are presented in the section on “EC cooling cycles”. We already stated that we are “assuming work recovery²¹”, but we see that we should explicitly state that we are “assuming 100% work recovery^{21,44}”, where new ref. 44 experimentally demonstrates 99% recovery. New ref. 44 is:

⁴⁴ S. Moench, R. Reiner, P. Waltereit, C. Molin, S. Gebhardt, D. Bach, R. Binniger, and K. Bartholomé, Enhancing Electrocaloric Heat Pump Performance by Over 99% Efficient Power Converters and Offset Fields. *IEEE Access* **10**, 46571-46588 (2022).

#7 Usually, it is believed that efficiency and COP is positively dependent on EC response (Nature Materials 13 (5), 439-450 (2014); Nature 600 (7890), 664-669 (2021)). This is

why large EC effect under low fields is required for the design of cooling devices. However, the authors show that “This increase of efficiency arises due to the small concomitant reduction in the magnitude of the EC effect (Fig. 4),”. More explanation should be provided on enhanced efficiency while reducing EC response.

The explanation was provided in Supplementary Note 19, now Supplementary Note 18. This Note is entitled “Increased efficiency with reduced EC effects”, and it was cited in the main paper for exactly this reason. The citing text has been simplified in our revised manuscript, and we now write:

“If MLCs of PST-PMW replace MLCs of PST (ref. 5) then prototypes¹⁻⁴ would operate with slightly improved efficiencies (Fig. 5f) due to small concomitant reductions in the EC effect (Fig. 4), as explained in Supplementary Note 18”.

Note that the aforementioned simplification has led to old Supplementary Note 20 becoming new Supplementary Note 18, such that old Notes 18,19 have become new Notes 19,20.

#8 In previous Nature work (Nature 575, 468 (2019) or ref. 5), the authors have made a comparison of EC effect with magnetocaloric effect in Gd under different temperatures to demonstrate the advance of EC multilayer capacitors for solid-state cooling applications. Six years later, the results on Gd are still used for comparison. Note that magnetocaloric effect in Gd has been well studied since 1976 (G. V. Brown. J. Appl. Phys., 47,3673, (1976)). There have been many advances in magnetocaloric materials, especially during the past two decades which show better magnetocaloric response than Gd (Progress in Materials Science 93, 2018, 112-232). Many prototype devices have been fabricated based on various magneocaloric materials (Kitanovski, A. et al. (2015). Overview of Existing Magnetocaloric Prototype Devices. In: Magnetocaloric Energy Conversion. Green Energy and Technology.). It is therefore suggested to add the comparison with the state-of-the-art magnetocaloric response instead of Gd.

We addressed this comment earlier – see above, at the place where we inserted the new figure for new Supplementary Note 15.

#9 “Like MLCs of PST (ref. 5), supercritical fields yield large EC effects over a wide range of temperatures”.

Supercritical phase transition has been used in PST reported in previous work (ref. 5) to rationalize the large EC effect in PST under high fields. What is the unique feature in PST-PMW whereas PMW may tune the supercritical phase transition behavior. If there is not, the inclusion of PMW shifts the transition temperature towards lower temperature.

We cannot understand the question in this comment. We explained that:

“we have developed MLCs in which the partial substitution of PST with PMW... disrupts dipolar order while preserving crystallographic order, resulting in first-order transitions with reduced Curie temperatures”

... and our text quoted by the referee alerts the reader to the fact that we drive the first-order transition supercritically.

As PST and PMW have both extensively studied during the past 6 years, the combination of PST and PMW is not considered as a breakthrough to the field even though EC effect has not been studied in PST-PMW. This lacks sufficient conceptual advance as required by the high standard in Nature.

Solid-solution PST-PMW has not been studied. We have therefore studied a novel composition. Even if PST-PMW had been studied in some way, one cannot say that all future work on PST-PMW would necessarily not be novel.

Additionally, EC studies often involve solid solutions that dilute ferroelectrics with paraelectrics (e.g. BaTiO₃-SrTiO₃ and BaTiO₃-BaZrO₃), but the intended suppression of operating temperatures is accompanied by an unintended suppression of EC effects. Our strategy of forming a solid solution between ferroelectric and antiferroelectric materials might be novel, and to recognise this, our final sentence now has the relevant information added to the existing parenthetical text, such that it reads:

“our materials development strategy (valence size sintering, ferroelectric-antiferroelectric solid solution) should inspire improvements in other ceramic materials, for EC applications and beyond.”

#10 The partial substitution disrupts dipolar order while preserving crystallographic order, resulting in first-order transitions with reduced Curie temperatures of $T_C \sim 230$ K (85PST-15PMW) and $T_C \sim 242$ K (90PST-10PMW) measured on heating (Supplementary Note 4).

The thermally driven zero-field entropy change for the transition of $|\Delta S_0| = 13.4$ kJ K⁻¹ m⁻³ in 85PST-15PMW which is smaller than $|\Delta S_0| = 14.8$ kJ K⁻¹ m⁻³ obtained in 90PST-10PMW. This indicates the smearing of the phase transition near T_C . However, the thermal hysteresis increases from 10 K to 14 K as PMW concentration increases from 10 PMW to 15 PMW. What is the origin of increased thermal hysteresis here?

The small increase in thermal hysteresis on increasing PMW content is due to increased disorder. This is interesting to explain in the paper. Therefore we have:

- Changed this text in the Discussion and outlook section:
“The partial substitution disrupts dipolar order while preserving crystallographic order, resulting in first-order transitions with reduced Curie temperatures of $T_C \sim 230$ K (85PST-15PMW) and $T_C \sim 242$ K (90PST-10PMW) measured on heating (Supplementary Note 4).”
to:
“The partial substitution preserves crystallographic order, but disrupts dipolar order, thus reducing transition temperature and increasing thermal hysteresis (Supplementary Note 4 and Table S6 therein).”

... where deleting the T_C values represents an improvement because there was no need to repeat them in the “Discussion and outlook” section.

- Made concomitant additions to Supplementary Note 4 by:
 - Writing this in the main text of the Note:

“From Fig. S5a,c, we see that increasing PMW content, and thus disorder, reduces absolute transition temperatures and increases thermal hysteresis (Table S6).”

- Adding new Table S6:

Composition	Transition temperature on heating (K)	Thermal hysteresis (K)
PST	292	4
90PST-10PMW	242	10
85PST-15PMW	230	14

Table S6. Variation of transition temperature and thermal hysteresis in PST due to PMW doping. Data from ref. S5 (PST) and Fig. S5a,c (PST-PMW).

#11 PST or PST-PMW contain toxic lead while a recent work on lead-free high-polar-entropy ceramics exhibits higher EC-induced temperature change near room temperature (Nature 640, 924930 (2025)). For instance, an EC temperature change of 10 K is achieved at an electric field of 10 V/μm near room temperature which is 3 times as large as that obtained by lead-contained PST-PMW under a higher electric field of 21.4 V/μm. EC response below room temperature was also studied in this previous work (Nature 640, 924930 (2025)). Moreover, MLCs based on lead-free ceramics have also been fabricated (Nature 640, 924930 (2025)). It is suggested to add a direct comparison of the magnitude of EC response near room temperature between current work and previous results (Nature 640, 924930 (2025)) to address the main advance over this recent work in Nature.

The *Nature* 2025 work does not demonstrate an EC working body that could be used for cooling through room temperature: the sample tested above and below room temperature is a thin single layer of thickness ~100 μm, while the MLC was tested at room temperature only and is compromised by substantial inactive thermal mass. Our revised paragraph 2 now explains about both of these samples and cites *Nature* 2025 as ref. 34 in the two relevant places:

“EC effects in existing material systems cannot be exploited for cooling through room temperature. For PST modified via B-site disorder³² or solid solution³³, sub-room-temperature EC effects were reported in thin single layers (~100 μm) and are small (≤1.6 K). For other ceramics, sub-room-temperature EC effects are sometimes larger (≤1-4 K), but they arise in thin single layers³⁴, are compromised by substantial inactive thermal mass³⁵, or decay to zero and change sign on crossing room temperature (as seen for antiferroelectric PMW)³⁵⁻³⁷. (Ref. ³⁴ also demonstrated an MLC that is compromised by substantial inactive thermal mass, and was only measured at room temperature.)”

Referee #3

We thank the referee for the detailed feedback, which we have used to improve the manuscript.

All changes (except minor figure changes) are shown using **yellow** highlights, or by highlighting a word in **blue** to indicate that all subsequent text in the paragraph is substantially or wholly new - this avoids the need to read large chunks of highlighted text.

These changes include changes due to feedback from all referees, changes where we found ways to improve the presentation, and changes that we made to comply with the Editorial advice to shorten the paper. Note that the improved presentation has led us to swap the order of Supplementary Notes 18-20.

Minor figure changes are as follows:

- In Fig. 4, using the feedback of referee 2, we have extended PST data below the Curie temperature using indirect data in order to show that PST performs well only above room temperature. These indirect data are shaded pale, and so the other data are no longer presented with (avoidable) variable shading (we have likewise removed variable shading from Fig. S17). We also used the feedback of referee 2 to repeat Fig. 4 in new Supplementary Note 15 and add data for other state-of-the-art magnetocaloric materials that are being used in prototypes.
- In Fig. 5f, we have improved the way we present the shading and made a proper scale bar. Equivalent changes have been made in what is now Fig. S29 (whose panels have been swapped).

Also, at the request of the Editor, the title has been shortened, and the two Tables have been moved to Extended Data.

Key results: Guo et al. report on a solid solution of PST and PMW which show electrocaloric effects on the order of several Kelvin over a broad range of temperature, around, and including room temperature. The MLCCs are robust against field fatigue ($>10^7$ cycles) and can in principle show high efficiencies (70-90%) when incorporated into regenerator coolers. This new composition significantly improves the temperature span of regenerator coolers, compared to PST, while also being more straightforward and less energetically costly to manufacture (long annealing steps required for PST are obviated). Comprehensive electrical and thermal characterization of the samples has been carried out to benchmark performance with good agreement between indirect electrical and direct temperature measurements being obtained, therefore supporting the authors conclusions about the electrocaloric performance of PST-PMW MLCCs.

Originality and significance: The fact that this new composition significantly improves the temperature span of regenerator coolers while also being more straightforward and less energetically costly to manufacture (long annealing steps required for PST are obviated) represents a substantial step forward for electrocalorics research. PST has been considered to be the best electrocaloric material and PST MLCCs have been used in the leading electrocaloric cooler prototypes (Wang et al. Science 370 370, 129 2020; Li

et al. Science 382, 801, 2023; Metzdorf et al. Commun. Eng. 3, 55, 2024). While there is a trade off in diluting PST with PMW in terms of magnitude of EC effect, this is made up for by the increased operating span across room temperature, making it very promising for applications. As the author's point out, there is still room for further optimization based on the strategy of valence-size sintering that has been outlined here.

Clarity and context: the text is well written, succinct and clear to read. I think it is accessible for a generalist audience and communicates the important points well. The references cited a suitable and provide a fair reflection of the state of the art.

We thank the referee for these positive comments.

Queries / Suggested improvements:

The authors outline their choice for choosing PMW to alloy with PST on the basis of comparable B-site cation dissimilarity and larger valence disparity. Since this design principle will be of great interest to readers, it would be useful for readers if they could suggest other candidate solid solutions that might be worth exploring.

Our final sentence provides design ideas by stating that:

“our materials development strategy (valence-size-sintering, ferroelectric-antiferroelectric solid solution) should inspire improvements in other ceramic materials, for EC applications and beyond”.

... with the underlined text added following the feedback of Referee 2.

Readers would indeed like to learn specific details about possible future work, but we do not think they would expect us to provide such details and lose our advantage.

How have the authors decided upon the chosen solid solution compositions of 10% and 15% PMW? This does not seem to be discussed in the paper or a framework for how the proportion of dilution by PMW is expected to tune the electrocaloric response and temperature span. It would be interesting to hear the authors' expectation for this and if they have been prevented from further exploration of the compositional phase space due to practicalities or challenges faced with sample fabrication

To explain this, we have revised what is now paragraph 3 of the main text. The relevant part reads:

“Here we describe large-active-volume MLCs based on hitherto unexplored solid solutions of $(1 - x)\text{PST}-x\text{PMW}$. Preliminary screening in $0.05 \leq x \leq 0.25$ revealed $dT_C/dx \sim -320$ K without EC performance degradation. We selected $x = 0.15$ (85PST-15PMW) and $x = 0.10$ (90PST-10PMW) to achieve EC effects in a range that runs from relatively low temperature (240 K) to well above room temperature.”

This revision has also improved the manuscript because we now present and define our two compositions in the introduction rather than just after the introduction, which in retrospect we realise was too late.

The authors suggest a relaxor-like behavior for PST-PMW supported by dielectric dispersion and well-established behavior for PST. Do the authors have any idea about the microstructure/polar configuration in the relaxor-like phase and how this is affected by PMW, which prefers antiferroelectric ordering (Li et al Adv. Func. Mater. 31, 2101176, 2021)? More of an explanation of the microscopic basis for the order-disorder entropy changes underpinning the observed EC effects would be useful or some clarification if the authors expect the effects to be primarily derived from latent heat as in the report for PST (Nair et al Nature 575, 468, 2019). In this regard, is there scope for discussion on the relative role of compositional disorder (B site cation distribution) versus dipolar entropy contributions, such as in the case of high polar entropy perovskites? (e.g. Du et al Nature 620, 924, 2025).

We explained in the main paper that:

“Indirectly measured values of $|Q|$ (solid lines, Fig. 3b) match well with the directly measured values, a weak relaxor behaviour notwithstanding (Supplementary Note 12)”.

However, the comment (and a related comment from Referee 2) reveals that our text did not make it clear that the match means that the weak relaxor behaviour may thus be ignored.

We have therefore changed the text quoted above to read:

“Indirectly measured values of $|Q|$ (solid lines, Fig. 3b) match well with the directly measured values, such that weak relaxor behaviour (Supplementary Note 12) may be ignored.”

We also now explain why the relaxor behaviour is weak by writing in Supplementary Note 12 that “The weak relaxor behaviour arises from the weak disorder in our weak solid solutions.”

Given negligible relaxor behaviour, we do indeed “expect the effects to be primarily derived from latent heat as in the report for PST (Nair et al Nature 575, 468, 2019)”, and as stated in the original manuscript, we “supercritically drive a first order ferroelectric phase transition above a low Curie temperature”. The statement in our revised manuscript includes more information, and reads:

“The operating temperature range is wide because a large electric field supercritically drives the first-order ferroelectric phase transition above a Curie temperature that is suppressed due to PMW disruption of PST dipolar order.”

The robustness against fatigue over the course of 10^7 cycles at 600 V is impressive but have the authors determined what is the actual breakdown field for these samples and how it compares to other high performance electrocalorics prepared in MLCC format? If the breakdown field is superior to PST then there is perhaps scope to mitigate the lower EC temperatures achieved by driving the MLCC with larger fields.

We thank the referee for recognising the impressive breakdown strength. We selected 600 V to match the voltage that can be repeatedly applied to MLCs of PST in prototypes (refs 1-4),

and so we only mention in passing our results for 750 V (Supplementary Note 14). We do not anticipate that our MLCs of PST-PMW can be reliably operated for many cycles above 600 V, but breakdown field magnitude and standard deviation could be improved if the MLCs were fabricated via semi-automatic or automatic production.

In previous works the authors have used IR camera imaging to directly map temperature fields but here they have used only thermocouple measurement. I appreciate this is challenging for below room temperature but could be still be used for independent room temperature direct measurement verification. This would add further confidence to the reported EC temperature changes and reveal the spatial distribution of temperature across the MLCC devices.

Indeed, our original manuscript stated that “infrared imaging is challenging below room temperature”.

However, our thermocouple measurements are safe because the temperature changes that we report are consistent with our indirect measurements (Fig. 3d), and these indirect measurements are consistent with our isothermal heat measurements (Fig. 3b).

The good match between thermocouple and IR data in previous work by a partially overlapping set of authors (ref. 5) also demonstrates that such thermocouple data from the Cambridge laboratory is reliable.

Spatial distributions of temperature should be similar to those observed for MLCs of PST (right inset of Fig. 3a in ref. 5, reproduced below). The infrared image shows the MLC very soon after removing the driving voltage, such that most of the active area (within black line) is at its coldest. The image shows that heat has leaked from the inactive area (outside black line) into the periphery of the active area, as one would expect.

Referee #4

I co-reviewed this manuscript with one of the reviewers who provided the listed reports.

We thank the referee for the effort.

Referee #5

I co-reviewed this manuscript with one of the reviewers who provided the listed reports.

We thank the referee for the effort.

Referee #1 (Remarks to the Author):

The questions and concerns have been well addressed. Given the significance of the findings and the solid theoretical research, we are pleased to recommend this work for publication in *Nature*. We believe that it will make a significant contribution to the field.

We thank the Referee for the positive comments and the recommendation to publish in *Nature*. Further changes are described in our response to Referee 2.

Referee #2 (Remarks to the Author):

The referee would like to thank the authors for their efforts in revising the manuscript and for addressing the previous comments. However, the additional arguments/corrections don't mitigate the concerns raised by referee 2 including the novelty of the work, the involved physical mechanism or the practical breakthrough of the current work, even after revision. Besides, this manuscript also contains various inconsistencies between the measured EC responses and the theoretical understandings. Therefore, the referee 2 cannot recommend this work for publication in *Nature*.

Referees 1, 3 and 4 recommend publication in *Nature*. We thank Referee 2 for the further feedback, which is very long and must have taken significant time. We really appreciate this, especially because we have used the feedback to make substantial improvements to the paper:

- We have introduced a new Supplementary Note 1 to explain why it is thermodynamically impossible to use PMW to cool a load through room temperature, such that it is now clear that no existing EC materials can be used to cool a load through room temperature. Other candidates are thin single layers, which do not possess sufficient thermal mass to be useful, and the EC effects are relatively small.
- We now summarize our key materials advance all together, both in our revised abstract where we write:

“Here we show that exaggerating valence mismatch via dilution with $\text{PbMg}_{0.5}\text{W}_{0.5}\text{O}_3$ (PMW) maintains high B-site order and latent heat with no anneal, while disrupting dipolar order to reduce the Curie temperature to 230 K. Our MLCs of PST-PMW display supercritical EC effects of ~ 3 K across and well below room temperature”

and at the start of a revised paragraph 3 in the main text, where we write:

“Here we describe large-active-volume unannealed MLCs based on hitherto unexplored solid solutions of $(1-x)$ PST- x PMW, where PMW disruption of PST dipolar order suppresses T_C without overly disrupting B-site order, such that the latent heat of the ferroelectric phase transition remains large³⁸. Supercritically driving this transition yields large EC effects over a wide range of operating temperatures that extends above and well below room temperature.”.

- The start of the Discussion section provides a complementary summary of our materials advance.
- We now clarify in the manuscript that our discussion of EC cycles provides “limiting upper bounds on performance” given that we assume ideal regeneration and work recovery.
- We no longer describe the relaxor behaviour as weak. In fact, we cannot establish the strength of the relaxor behaviour, as we cannot acquire dielectric spectra at much higher frequencies given self-resonance in our large MLCs (now shown in Supplementary Note 5).
- We now discuss the zero-field transition before presenting EC measurements, and we see that thermal hysteresis in the heat capacity data matches thermal hysteresis in fresh dielectric spectra, where we now show cooling data as well as heating data.

In addition to the changes to the text that we discuss below in our point-by-point response to the detailed comments of Referee 2, and a very small number of very minor other changes, we have improved the first paragraph on EC measurements, where to avoid an ambiguity associated with field application and field removal, we now use T_s to mean set temperature rather than starting temperature.

All changes to the text are shown using **yellow** highlights, or by highlighting a word in **blue** to indicate that all subsequent text in the paragraph is substantially or wholly new – this avoids the need to read large chunks of highlighted text. Additionally, the titles of Supplementary Notes 1 and 5 are highlighted blue to indicate that the entire Note is new (Note 1) or heavily modified (Note 5). However, for simplicity, we have not marked changes that arose from the renumbering of Supplementary Notes, Supplementary Figures, and Supplementary References.

Figure changes are as follows:

- New Supplementary Note 1 contains three new figures (Figs S1-3).

- New Fig. S8 (old Fig. S14) now shows new and less noisy dielectric spectra on cooling as well as heating. Note that we used fresh samples for operational reasons; more details on this are given below in our response to the relevant point.
- Fig. S9 has been added to show that higher frequency dielectric spectra would be compromised by MLC self-resonance.
- Colour maps in Fig. 5c-e and Figs S30-32 (formerly Figs 25-27) now have horizontal axes that do not overlap with data for which $T_h - T_c = 0$ and 30, and maps showing COP data now have correctly coded colours and scalebars.

First, the claim that “EC effects in existing material systems cannot be exploited for cooling through room temperature.” is not fully correct as there are existing materials that have been developed for cooling through room temperature. To make a rational comparison of the EC properties, it is always useful to show the data deduced under the same electric field. This is crucial especially for the fact that PST-PMW layer in MLCC is much thinner than that in bulk ceramics which allows much higher electric field to be applied to drive higher EC response. In this regard, the referee mention here that they have for instance cited the modified PST ceramics reported in (R. Yin et al., J. Adv. Ceram. 14, 9221088 (2025)) which finally exhibits slightly higher EC-induced effect than that of PST-PMW (proposed in the current work) when considering the same electric field (i.e. 6 V/ μm). Meanwhile, the authors also recognized that PST-based solution exhibit comparable EC response to current work under the same electric field according to “Correcting 1.8 K for layer thermalization using the factor of 0.85 that we give in the main paper implies an effective temperature change of $|\Delta T_{\text{eff}}| \sim 1.5$ K, which is similar to the value quoted by the referee.” It is not fully fair to impose the comment that these results are “below 1.6 K”. The reason why PST-PMW achieves higher EC response than 1.6 K is the much higher electric field used in MLCC thanks to thinner thicknesses. As a matter of fact, ECE deduced from PST-PMW is not enhanced compared with PST solutions reported 33 years ago (Ferroelectrics 127, 143 (1992)) and state-of-the-art results in modified PST (J. Adv. Ceram. 14, 9221088 (2025)) without the need for long-term annealing and/or complex composition.

As explained in paragraph 2 of the main text, EC cooling through room temperature is not possible because “existing materials that have been developed for cooling through room temperature” cannot be used: one either has insufficient thermal mass or else one has unsuitable EC effects:

- The EC materials with insufficient thermal mass are “thin single layers”, and nobody is making prototypes with such materials because the use of such thin layers would be impractical.
- The unsuitable EC effects arise in PMW. We previously stated that they are unsuitable because they “decay to zero and change sign on crossing room temperature”. We now explain in a detailed new Supplementary Note (Note 1) exactly why it is thermodynamically impossible to use PMW to cool to or through room temperature.

The comparison made by the Referee with “existing materials that have been developed for cooling through room temperature” is therefore moot, but in response:

- Comparing data obtained with the same field can indeed be instructive, but for cooling it is important to maximise EC effects by using as large a field as possible.
- Consequently, the many thin layers in MLCs represent an unmitigated advantage.
- Nevertheless, if we examine the Referee’s comparison of EC effects at $6 \text{ V } \mu\text{m}^{-1}$:
 - Ref. 32 [R. Yin et al., *J. Adv. Ceram.* 14, 9221088 (2025)] reports 1.6 K for PST with maximum disorder ($S_{111} = 0.51$) at around 273 K [Fig. 5(a) in ref. 32].
 - We report 2.5 K for 85PST-15PMW at 246 K (Fig. 2c).

The Referee has nevertheless stated that EC effects in ref. 32 are “slightly higher”. Perhaps the Referee looked at EC data for samples with higher B-site order, but these EC effects arise above room temperature and are therefore irrelevant here.

- Given the need to maximise EC effects by using as large a field as possible, it is indeed “fair” to compare our results with values “below 1.6 K” that were achieved when only $6 \text{ V } \mu\text{m}^{-1}$ could be applied, but even if we compare at $6 \text{ V } \mu\text{m}^{-1}$ then we have the improvement described in the previous bullet point.
- Our EC effects of $\sim 3 \text{ K}$ exceed those that fall “below 1.6 K” in the two references to which the Referee refers (refs 32 and 33 in paragraph two of our main text). It is therefore unreasonable to say we have not improved on those works. Not only do we have larger temperature changes (even at the same field, let alone our higher fields), we have macroscopic working bodies that can be used to cool a load through room temperature.
- The Referee states at the end of the comment that two papers (our refs 33 and 32, respectively) did not require “long-term annealing and/or complex composition”. However, it is incorrect to imply that our unannealed novel solid

solutions (PST-PMW) are any more complex than the simple solid-solutions described in ref. 33, and it is incorrect to imply that ref. 32 did not require long-term annealing because the PST samples were annealed once or twice for 190 hours at 900 °C.

Please note that in paragraph 2, we have deleted the claim that MLCs of PMW display “substantial inactive thermal mass” – we are sorry to have written this, and we would like to explain that we made the mistake because we compared inactive outer layer thickness with individual active layer thickness rather than total active layer thickness.

I our previous comment, the referee mentioned that “The smaller EC strength in current work than previous one strongly declines the advance of current work given that EC strength of EC materials is crucial to practical cooling applications.” The authors further admitted that the EC strength is not enhanced through PST-PMW solid solution according to their sentence “After correcting the referee’s estimate of 0.8 K to 1.5 K two answers ago, one finds that the two EC strengths are equivalent.”. This is also recognized by reviewer 3 that expanding the operating below room temperature is achieved at the expense of reduced EC response and EC strength. However, the authors said that “EC strength is of secondary importance and does not affect our main claim to have demonstrated an EC working body that could be used to cool a load through room temperature.” As the referee already mentioned in their previous review, this work does not represent the first example demonstrating “EC working body that could be used to cool a load through room temperature”. Please refer to the recent work on EC cooling through room temperature based on PMW MLCC (R. Yin et al., Adv. Funct. Mater. 2502550 (2025)) published before the submission of current work.

The Referee describes the previous correspondence in which we explained that EC strength is of secondary importance, and that our main claim is to have demonstrated an EC working body that could be used to cool a load through room temperature. The Referee goes on to dispute our main claim by claiming that R. Yin et al., Adv. Funct. Mater. 2502550 (2025) reports MLCs of PMW that would permit cooling through room temperature. However, cooling through room temperature is not claimed in that paper, and as we explained in our manuscript, PMW cannot be used to cool a load through room temperature because the EC effects “decay to zero and change sign on crossing room temperature”. We had felt that this explanation was sufficient, but we now explain why PMW cannot be used to cool a load through room temperature by presenting cooling cycles in a new Supplementary Note 1 (four sides, three figures). In the main

text, we now write that the decay to zero and change of sign make it “thermodynamically impossible to use PMW to cool to or through room temperature (Supplementary Note 1)”.

Note that an overlapping set of the present authors reported MLCs of PMW [Sakyo Hirose et al 2023 J. Phys. Energy 5 035009] prior to the 2025 paper discussed above. In that paper, we suggested that MLC of PMWs could be used for cooling *near* room temperature like MLCs of PST [*Nature* 575 (2019) 468-472], but we did not suggest that PMW could be used for cooling *through* room temperature. This is because we knew that EC effects falling to zero and changing sign near room temperature would make cooling through room temperature thermodynamically impossible, as we now explain via cooling cycles in Supplementary Note 1.

Second, although the authors tried to comment on previous works, the significant challenge why the previous works failed to achieve “EC effects exploited for cooling through room temperature” remains not clear to readers. Especially, the referee sees no key scientific mechanism preventing the practical use of EC ceramics and MLCC from cooling through room temperature that has been successfully overcome by the authors. From material’s perspective, EC effect below room temperature has been demonstrated with comparable EC value 33 years ago (Ferroelectrics 127, 143 (1992)). Indeed, the authors also recognized that “We are not claiming that the solid-solution method is novel, but please note that solid solutions of PST and PMW are “hitherto unexplored”.” PMW with larger enthalpy change than PST exhibit outstanding EC properties and cooling performance (COP, heat flux) is only mentioned with a short comment “decay to zero and change sign on crossing room temperature (as seen for antiferroelectric PMW)³⁵⁻³⁷”.

Paragraph 2, now citing the new Supplementary Note 1, explains that existing EC materials cannot cool a load through room temperature as either (1) they exist as “thin single layers” or (2) the EC effects “decay to zero and change sign on crossing room temperature (as seen for antiferroelectric PMW)³⁵⁻³⁷ making it thermodynamically impossible to use PMW to cool to or through room temperature (Supplementary Note 1).”

In Ferroelectrics 127, 143 (1992), large EC effects in some of the samples arise either wholly above room temperature (Fig. 1 of the 1992 paper) or wholly below room temperature (Fig. 2 of the 1992 paper). We improve upon this and all other previous

work because our MLCs of PST-PMW display larger EC effects over a wide range of temperatures that extends above and well below room temperature, and we do not need to anneal our MLCs. Our advance was achieved by employing partial B-site substitutions that exaggerate valence and are compatible in terms of size, such that we disrupt dipolar order without overly disrupting B-site order, thus reducing Curie temperature without unduly reducing latent heat.

We are sorry to see that this information was not clearly presented in any one place within the manuscript, and for this we apologise. To rectify the matter, we now explain in the abstract that:

“Here we show that exaggerating valence mismatch via dilution with $\text{PbMg}_{0.5}\text{W}_{0.5}\text{O}_3$ (PMW) maintains B-site order and thus latent heat with no anneal, while disrupting dipolar order to reduce the Curie temperature to 230 K. Our MLCs of PST-PMW display supercritical EC effects of ~ 3 K across and well below room temperature”

... and in paragraph 3 of the main text that:

“Here we describe large-active-volume unannealed MLCs based on hitherto unexplored solid solutions of $(1-x)\text{PST}-x\text{PMW}$, where PMW disruption of PST dipolar order suppresses T_C without overly disrupting B-site order, such that the latent heat of the ferroelectric phase transition remains large³⁸. Supercritically driving this transition yields large EC effects over a wide range of operating temperatures that extends above and well below room temperature.”

... and in the discussion that:

“The preservation of B-site order limits the suppression of latent heat, while the disruption of dipolar order reduces the Curie temperature above which large EC effects can be supercritically driven over a wide range of temperatures.”.

In addition, the mechanism for achieving EC response below room temperature proposed by the authors is that they “supercritically drive a first order ferroelectric phase transition above a low Curie temperature”, which is not new. Both supercritical phase transition and tuning of phase transition temperature using solution method were already proposed in the field, as also recognized by authors. Moreover, both EC response and EC strength is considerably reduced in the PST-PMW samples of the authors due to including PMW. Consequently, the EC results deducted from PST-PMW only make incremental contribution to the

field at the material level while outstanding scientific understanding is not revealed in the current work. In addition, even the current mechanism to explain EC behavior is questionable as the referee already found the existence of serious inconsistencies in characterizing the phase transition which fails to provide understanding of observed EC properties.

Our previous response explains what is new, and how the information is now more clearly presented in the manuscript. From our responses to all points above, what is new should now be clear to the Referee, who should therefore now appreciate that our work is not incremental. Instead, we have the first macroscopic EC samples that operate through room temperature, and could thus cool a load through room temperature, such that they should be used to replace MLCs of PST in existing and future EC prototypes in order to achieve EC refrigeration for the first time. The mechanism is also discussed above, while an inconsistency regarding the temperature of the peak in each dielectric spectrum is explained below and resolved.

Given the lack of sufficient conceptual breakthroughs, to meet the high standard required by Nature Journal, there must be striking advance for practical application in this work compared with previous studies. Unfortunately, the referee has already stated that the results obtained in PST-PMW are not sufficiently striking compared with the state-of-the-art MLCC based on PMW (R. Yin et al., Adv. Funct. Mater. 2502550 (2025)). The referee's point is that both PST-PMW and PMW have advantages and disadvantages which are summarized as follows. PST-PMW shows broad temperature span while the counterpart of PMW is narrow; The magnitude of largest EC response in PMW considerably outperforms that of PST-PMW under the same electric field, corresponding to higher EC strength; Processing of PST-PMW is more complex and costly than PMW simply due to complex solid solution; Ag/Pd electrode was used in PMW while much more expensive Pt electrode (the price is about 3 times as that of Ag and roughly the same as Pd) was used; PMW uses an oscillation circuit designed to recover energy while PST-PMW simply assumes 100% energy recovery without any circuits; COP based on PMW is 350 while COP based PST-PMW is about 100; The heat flux of PMW is 245 W cm⁻² while it is about 1.4 W cm⁻² for PST-PMW (77.7 mW; area 5.6 mm²); The driving voltage for PMW is 350 V while it is 600 V for PST-PMW. According to the summary above, there is no key evidence supporting that PST-PMW is more favorable than PMW except the broader temperature span achieved in PST-PMW compared with PMW. In other words, PST-PMW fails to demonstrate striking advance over PMW for practical cooling

applications. Even the authors recognized the promise of PMW in their previous paper that “This good EC performance near room temperature implies that MLCs of PMW could be exploited in prototype EC coolers.” (Sakyo Hirose et al 2023 J. Phys. Energy 5 035009). In addition, PST-PMW exhibits a cooling power of 77.7 mW assuming 100% energy recovery which is smaller than that (90 mW) obtained in real prototype based on PST MLCC (Y. Wang et al Science 2020, 370, 129) weakening the claim that “MLCs of PST-PMW should now replace MLCs of PST”.

As explained earlier, the conceptual breakthrough is that we modify PST by making partial B-site substitutions that exaggerate valence and are compatible in terms of size, we disrupt dipolar order without overly disrupting B-site order, thus reducing the Curie temperature without unduly reducing latent heat. There is also a conceptual breakthrough in terms of materials processing because we have avoided the anneal, as discussed in paragraph 2 of our Discussion section.

As also explained earlier, PMW cannot cool a load through room temperature.

As also explained earlier, EC strength is not a key parameter because, for cooling, it is important to maximise EC effects by using as large a field as possible.

The next comments by the Referee compare PST-PMW and PMW in terms of processing procedure, materials costs, energy recovery, COP, heat flux, and driving voltage. These comments are moot given that PMW cannot cool a load through room temperature. However, we note that the heat flux comparison between 400 MLCs of PMW and one MLC of PST-PMW is not reasonable.

Moreover, the comment on COP has led us to realise that our highest COP values were not apparent because (i) the highest marked values on our colour scalebars were too small, and (ii) data for $T_h - T_c = 0$ and 30 on the horizontal axis were obscured by overlap with the axis. We have therefore modified the colour maps in Fig. 5c-e and Figs S30-32 (formerly Figs 25-27). Specifically, horizontal axes no longer overlap with data for which $T_h - T_c = 0$ and 30, and maps showing COP data now have correctly coded colours and scalebars. The largest COP we show in Fig. 5e is 182, but we exceed the PMW value of 350 if we move to lower voltages. Specifically, for 300 V, we have 3658 for 85PST-15PMW (new Fig. S30c) and 991 for 90PST-10PMW (new Fig. S31c).

As stated earlier, the previous work coauthored by an overlapping set of the present authors [Sakyo Hirose et al 2023 J. Phys. Energy 5 035009] does indeed suggest that

MLCs of PMW could be used for cooling *near* room temperature like MLCs of PST [*Nature* 575 (2019) 468-472], but it does not suggest that PMW could be used for cooling *through* room temperature. This is because we knew that EC effects falling to zero and changing sign near room temperature would make cooling through room temperature thermodynamically impossible, as we now explain using cooling cycles in Supplementary Note 1.

Our statement that “MLCs of PST-PMW should now replace MLCs of PST” cannot be invalidated by comparing the performance of an MLC of PST-PMW with the performance of a prototype (based on nine MLCs of PST). Our statement is made on the basis of the like-for-like MLC comparison that appears in Fig. 4. Replacing MLCs of PST with MLCs of PST-PMW in prototypes would permit EC cooling through room temperature for the first time.

Nevertheless, the authors have added a new comment to distinguish the current work from previous works on PMW that “decay to zero and change sign on crossing room temperature (as seen for antiferroelectric PMW)³⁵⁻³⁷”. This comment is not fully acceptable as explained below. The former “decay to zero” is not relevant to EC cooling using PMW below room temperature as EC response in PMW only decays remarkably to zero only below -20°C. For practical cooling applications, -20°C is enough to meet the operating temperature range for cooling food, beverages, the built environment, and medicine as mentioned in the Introduction part. On the other hand, the sign change in EC response is not a drawback preventing PMW from practical applications. As this point has also been recognized in their previous paper (Sakyo Hirose et al 2023 *J. Phys. Energy* 5 035009) saying that “Note that we do not follow the common practice of describing inverse EC effects as negative EC effects because absolute sign depends on whether field is applied or removed, and whether one considers a temperature change in the adiabatic limit or an entropy change in the isothermal limit.”. The sign change can be simply overcome by applying electric field for the temperature range with a negative sign while removing electric field for the temperature range with a positive sign. Meanwhile, the sign change is not an issue if PMW is used for a specific temperature range near room temperature from 20°C to 35°C (the built environment) or 0-10 °C or -20-0°C (food, beverages, and medicine). Therefore, these analyses strongly suggest that ““decay to zero and change sign on crossing room temperature (as seen for antiferroelectric PMW)³⁵⁻³⁷”” cannot be regarded as a key factor hampering PMW for cooling through room temperature not to mention that current authors also supported the use of PMW for practical cooling

applications (Sakyo Hirose et al 2023 J. Phys. Energy 5 035009).

As we now explain in detail, it is thermodynamically impossible for PMW to cool a load through room temperature (new Supplementary Note 1).

It is therefore a moot point that EC effects fall to zero well below room temperature in both PMW (“-20°C”) and PST-PMW (230 K and 242 K for our two compositions, i.e. -43°C and -31°C).

Our previous answer gives our response to the comment on the paper coauthored by an overlapping set of the present authors.

For further evaluation of practical cooling applications, EC cooling cycles under ideal conditions have been studied by assuming 100% energy recovery. The authors referred to the work reported in ref. 44 whereas 99.74% energy recovery is achieved in PMN combined with specifically designed electrical circuits. Note that COP is strongly dependent on the value of the energy recovery during the cooling cycles as also recognized by authors in their previous paper (Nature Communications 9, 1827 (2018)). Despite 99.74% energy recovery achieved in PMN, previous work by authors shows that 70-80% energy recovery is achieved in BTO-based MLCC which decreases to 65% for a prototype refrigerator with 24 such capacitors (Nature Communications 9, 1827 (2018)). Without explicit demonstration using electrical circuits, the use of 100% energy recovery in PST-PMW MLCC remains yet to be technically testified simply because the electrical circuits were not really done in current work. Otherwise, one can argue that modified PST (R. Yin et al., J. Adv. Ceram. 14, 9221088 (2025)) should now replace MLCs of PST in EC prototypes for efficient EC cooling through ambient even though modified PST ceramics were not made into MLCC as done in this work.

Energy recovery with electrocalorics was pioneered by some of the present authors in the paper cited by the referee [Nature Communications 9 (2018) 1827], which is our ref. 21. Achieving better energy recovery is a matter of designing the right electronics and control software. It is reasonable to assume 100% recovery here, when the as-quoted practical values of 99.74% can be achieved.

In any case, given that we state our assumption of 100% work recovery, readers should appreciate that we are describing a theoretical upper bound on COP values. To make this clear at the outset of our section on EC cycles, we have changed:

“Fig. 5 describes cooling cycles (Fig. 5a,b) in which one or more MLCs, driven

using ≤ 600 V, undergoes Brayton-like cycles ($1 \rightarrow 2 \rightarrow X \rightarrow 3 \rightarrow 4 \rightarrow Y \rightarrow 1$) that involve translation ($X \rightarrow 3$, $Y \rightarrow 1$) between the hot and cold ends of an ideal fluid regenerator at T_h and T_c (T_c differs from Curie temperature T_C).”

to:

“Here we consider limiting upper bounds on performance by describing cooling cycles (Fig. 5a,b) in which one or more MLCs, driven using ≤ 600 V and assuming 100% work recovery²¹ (>99% has been demonstrated⁴⁴), undergoes Brayton-like cycles ($1 \rightarrow 2 \rightarrow X \rightarrow 3 \rightarrow 4 \rightarrow Y \rightarrow 1$) that involve translation ($X \rightarrow 3$, $Y \rightarrow 1$) between the hot and cold ends of an ideal fluid regenerator at T_h and T_c (T_c differs from Curie temperature T_C).”.

Our exciting suggestion of using MLCs of PST-PMW in prototypes that would cool through room temperature is realistic given that there are already prototypes based on MLCs of PST (which operate above room temperature only). It is not reasonable to suggest that thin single layers of modified PST [R. Yin et al., *J. Adv. Ceram.* 14, 9221088 (2025)] could be used in prototypes, as the thermal mass is too small.

The value of COP under ideal condition is about 100 according to Fig. 5e. By contrast, an oscillation circuit was designed to effectively recover the electric energy based on PMW MLCC (R. Yin et al., *Adv. Funct. Mater.* 2502550 (2025)). The COP with an oscillation circuit is 350 which is nearly 3 times higher than that obtained by PST-PMW without any circuit to recover the energy. This result explicitly demonstrates the promising potential of PMW MLCC for cooling application which may be considered more favorable than PST-PMW not mentioning several other advantages inherent to PMW. In this work (R. Yin et al., *Adv. Funct. Mater.* 2502550 (2025)), the authors further show that the heat flux of prototype cooler based on with 400 PMW MLCCs is 245 W cm^{-2} which correspond to three orders of magnitude higher than the state-of-the-art current EC coolers based on PST MLCCs (Y. Wang et al *Science* 2020, 370, 129). Meanwhile, the voltage used to drive such high cooling performance is about 350 V corresponding to nearly half of that used in PST-PMW counterparts. This again supports the referee argument that enhanced EC strength of EC materials is crucial to cooling applications.

We discussed earlier that it is thermodynamically impossible to use PMW to cool a load through room temperature.

The Referee’s earlier comment on a COP of 350 for PMW alerted us to the fact that our largest COP values exceed this value, as discussed earlier.

The heat flux calculated by assuming the use of many MLCs of PMW depends trivially on the stacking arrangement, and would be reduced by losses if such MLCs were used in a prototype. It would be interesting to fabricate a prototype based on MLCs of PMW for comparison with the prototype based on MLCs of PST, but neither would be able to cool through room temperature, which is the point of our materials development.

MLCs of PST-PMW can also be operated using 350 V. If they are, then as stated earlier, the COPs exceed the value of 350 for PMW, as we have 3658 for 85PST-15PMW (new Fig. S30c) and 991 for 90PST-10PMW (new Fig. S31c). Valuing EC strength implies operating at low voltage, but this reduces EC effects, and as stated earlier, for cooling it is important to maximise EC effects by using as large a field as possible.

Regarding the practical cooling applications with respect to magnetocaloric counterparts, the referee has already commented that “In previous Nature work (Nature 575, 468 (2019) or ref. 5), the authors have made a comparison of EC effect with magnetocaloric effect in Gd under different temperatures to demonstrate the advance of EC multilayer capacitors for solid-state cooling applications. Six years later, the results on Gd are still used for comparison. Note that magnetocaloric effect in Gd has been well studied since 1976 (G. V. Brown. J. Appl. Phys., 47,3673, (1976)). There have been many advances in magnetocaloric materials, especially during the past two decades which show better magnetocaloric response than Gd (Progress in Materials Science 93, 2018, 112-232).” Now the authors add further evidence to support that Gd remains the best material for developing magnetocaloric prototypes. This may be true for the magnetic field around 1 T (Adv. Energy Mater. 2019, 9, 1901322). For the magnetic field exceeding 2 T, there are various magnetocaloric materials exhibiting higher caloric response than PST-PMW reported in current work (Adv. Energy Mater. 2019, 9, 1901322). One can comment on the costly large magnetic field of 2 T while the other can also comment on the high voltage of 600 V used here exceeding the safety voltage in laptop, cellphone and refrigerator not mentioning safety concern over wearable devices. The use of Pt electrodes is also costly. PST-PMW MLCC here is not a prototype which is the key working body. The high EC performance is pushed by high electric field close to breakdown field (750 V as claimed by authors). The EC-induced temperature drops below 1 K for the voltage below 100 V, which is much smaller than benchmark magnetocaloric Gd under 1 T. Therefore, to avoid potential overselling of electrocaloric cooling over magnetocaloric counterpart, it is reasonable to put the state-of-the-art results obtained by magnetocaloric

materials under 2 T (Adv. Energy Mater. 2019, 9, 1901322). This is fair to make rational comparisons between different caloric materials given that 2 T is accessible to labs and industries.

Some of us have worked on magnetocaloric materials for many years, most notably Xavier Moya.

Magnetocaloric materials that require 2 T are not used with permanent magnets, which we generously assume to be able to deliver 1.4 T in a volume large enough to be useful.

High voltage does not compromise safety if currents are limited. It is current that damages biological tissue.

Pt cost would need to be set in the context of a full economic evaluation in order to establish its significance.

The use of smaller voltages will reduce EC effects, but as we said earlier, for cooling it is important to maximise EC effects by using as large a field as possible.

We make a like-for-like comparison between the performance of our MLCs and the performance of magnetocaloric Gd bed, i.e. the embodiment of Gd used in prototypes with permanent magnets. Comparing with any magnetocaloric material under conditions that are not achieved in prototypes (e.g. using 2 T) is not meaningful.

Consequently, striking advances for practical applications are not revealed in current work compared with the state-of-the-art results in PMW MLCC which exhibits numerous advantages than PST-PMW solutions.

We refer the Referee to our previous response in which we explain that PMW cannot cool through room temperature because it is thermodynamically impossible.

Inconsistencies between the measured EC responses and the theoretical understandings

The points raised below are addressed below.

The mechanism for EC response in PST-PMW is also questionable. The DSC data is not consistent with dielectric results; the structural origin of increased disorder

remains out of reach and the correlation between local structure and first order phase transition remains elusive. These concerns are critically linked to EC properties which are highly relevant to main claim of this work, which has not been fully addressed or just ignored by authors.

The main claim of this work is that we have developed the first EC material systems that could be used to cool a load through room temperature. Other relevant issues are addressed in our responses below.

Both referee 2 and referee 3 raised concerns on the relaxor behavior. However, the authors insisted on negligible role of relaxor in EC response based on dielectric spectra under different frequencies and good agreement between direct and indirect results. The referee is not fully convinced by authors' response, which is explained as follows:

As explained in the responses below, we have made changes to what we say about relaxor behaviour in light of the feedback of the Referee.

First, it is early to conclude that relaxor behavior is weak. The referee 2 has already commented that “only three frequencies were shown in Fig. S14, which are not enough to analyze the relaxor behavior” while the authors argued that “The three frequencies differ enough to show that relaxor behaviour is weak and may be ignored.”. The strength of relaxor behavior can be analyzed by the shift in dielectric peak temperature ΔT_m between low and high frequencies (i.e. 100 Hz c 1 MHz in Nature Materials 17, 718–724 (2018)). Although ΔT_m between 100 Hz and 10 kHz is small, the shift between 100 Hz and 1 MHz can be big.

We no longer describe the relaxor behaviour as weak, so we have deleted “weak” in the main text, in the title of the relevant Supplementary Note (New Note 5, old Note 12) and at all relevant places in the text of Supplementary Information, i.e. everywhere.

Based on dielectric data with limited frequency range (Fig. S14), on can see: $T_m=252.2$ K at 100 Hz and $T_m= 254.9$ K at 10 kHz for 85PST-15PMW; $T_m=258.4$ K at 100 Hz, $T_m=259.5$ K at 1 kHz and $T_m= 261.4$ K at 10 kHz for 90PST-10PMW. ΔT_m between 100 Hz and 10 kHz is 2.7 K for 85PST-15PMW which is slightly smaller than that (3 K) for 90PST-10PMW. This result indicates that relaxor behavior is weakened with less disorder with increasing PMW content which contradicts with the claim that “The small increase in thermal hysteresis on

increasing PMW content is due to increased disorder.” The referee is also confused that the authors have already mentioned that “The weak relaxor behaviour arises from the weak disorder in our weak solid solutions”. Given that the weak relaxor behavior has been ignored, why is the increase in thermal hysteresis on increasing PMW attributed to increased disorder?

Let us address this comment with reference to new and less noisy dielectric data (new Fig. S8, reproduced below) that we collected on cooling as well as heating in light of the Referee’s next comment.

- These new data were obtained using a different measurement system, and so the Methods section on Dielectric permittivity has been revised accordingly.
- The fact that the original (different) measurement system was in Japan (the UK) has led us to measure different MLCs of each composition, such that each reference to MLC4 now describes an MLC that is different with respect to the corresponding MLC reported in our previous submission.

The as-marked peak temperatures for our lowest and highest frequencies reveal frequency shifts of $247.3 - 244.7 = 2.6$ K (85PST-15PMW) and $254.4 - 256.8 = 2.4$ K (90PST-10PMW). It was therefore correct, and not incorrect, to say that the composition with greater PMW doping shows a larger shift of dielectric peak temperature with frequency due to the greater disorder and thus stronger relaxor behaviour. (This was also true for our original data in old Fig. S14, but we appreciate that peak values are difficult to read from such plots, we apologise for not providing peak values last time, and we appreciate the fact that the Referee tried to identify them.)

As stated in our previous response, we no longer say that the relaxor behaviour (or the disorder) is weak. Therefore the Referee should now find our overall position to be self-consistent: higher PMW content implies greater disorder, leading to relaxor

behaviour in which the shift described above is enhanced, and larger thermal hysteresis due to increased pinning associated with the energy barriers between metastable states in a complex free-energy landscape (as discussed three responses ahead).

More importantly, the dielectric peak temperature- a signature of the phase transition temperature also contradicts with the results obtained by heat capacity measurement (Fig. S5). For instance, the dielectric peak temperature for 90PST-10PMW is around 260 K (Fig. S14) which is about 20 K higher than that obtained by heat capacity measurement (Fig. S5). Meanwhile, the dielectric peak temperature for 85PST-15PMW is around 252 K (Fig. S14), which is about 10 K higher than that obtained by heat capacity measurement (Fig. S5). Such large discrepancies in the phase transition temperature cannot be attributed to the difference arising from different experimental techniques.

We had not noticed that the peaks we showed in dielectric spectra (old Fig. S14) occurred above the temperatures at which the thermal hysteresis is observed in DSC (new Fig. S7a,c, old Fig. S5a,c).

To investigate, we obtained the new and less noisy dielectric spectra on both cooling and heating, as described and shown in our previous response.

The new dielectric spectra and the DSC data both show thermal hysteresis at similar temperatures. The peaks in the dielectric spectra lie at higher temperatures. We now explain via our revised Supplementary Note on relaxor behaviour (new Note 5, old Note 12) that:

“This coexistence between first-order behaviour and relaxor behaviour has been seen in other ceramics^{S12,S13}.”

[S12] Y. Yao, Z. Sun, Y. Ji, *et al.* Evolution of the tetragonal to rhombohedral transition in $(1 - x)(\text{Bi}_{1/2}\text{Na}_{1/2})\text{TiO}_3 - x\text{BaTiO}_3$ ($x \leq 7\%$). *Sci. Technol. Adv. Mater.* **14**, 035008 (2013).

[S13] E. V. Colla, N. K. Yushin, and D. Viehland, Dielectric properties of $(\text{PMN})_{(1-x)}(\text{PT})_x$ single crystals for various electrical and thermal histories. *J. Appl. Phys.* **83**, 3298 (1998).

The aforementioned improvements have led us to correct an error of omission, such that we now describe the zero-field thermally driven transition in the main text before

describing EC measurements. Specifically, we have added a paragraph that reads:

“The thermally hysteretic transition observed in heat capacity data reveals that increasing PMW content, and thus disorder, reduces absolute transition temperatures and increases thermal hysteresis (Supplementary Note 4). Dielectric spectra reveal that this thermal hysteresis coexists with a relaxor behaviour arising due to PMW disruption of PST (Supplementary Note 5).”

Abnormal behavior also occurs in magnitude of dielectric constant under different frequencies. The dielectric constant at 100 Hz and 1 kHz for 90PST-10PMW is higher than those for 85PST-15PMW. By contrast, the dielectric constant at 10 kHz for 90PST-10PMW is abnormally smaller than that in 85PST-15PMW.

The abnormality is not longer observed in our new and less noisy dielectric spectra – please see the figure inserted two responses back.

“The small increase in thermal hysteresis on increasing PMW content is due to increased disorder.” This explanation is also questionable. The increase in thermal hysteresis is usually indicative of strengthening of the phase transition or sharper phase transition. For instance, first order phase transition is accompanied by notable thermal hysteresis while second order phase transition exhibits weak or vanishing thermal hysteresis. The stronger phase transition implied from the increased thermal hysteresis usually results in stronger EC response in PST-PMW which contradicts with the main experimental results. The sharper phase transition caused by including PMW should give rise to higher dielectric constant with increasing PMW content, which also contradicts with dielectric spectra data (Fig. S14). The referee also refers the zero-field entropy change for the transition of $|\Delta S_0| = 13.4 \text{ kJ K}^{-1} \text{ m}^{-3}$ (85PST-15PMW) which is smaller than $|\Delta S_0| = 14.8 \text{ kJ K}^{-1} \text{ m}^{-3}$ (90PST-10PMW) in Fig. S5.

The assertion that a “stronger phase transition” should increase thermal hysteresis is incorrect given classic literature describing e.g. the C-T and O-T transitions in the archetypal ferroelectric BaTiO₃, and our data. We now explain the trend in our data by writing in Supplementary Note 4 that:

“disorder... increases thermal hysteresis due to increased pinning associated with the energy barriers between metastable states in a complex free-energy landscape”.

Now that we present dielectric spectra measured on both heating and cooling (see the figure presented earlier), it can be seen that the peaks in the dielectric spectra lie wholly

above the temperatures at which there is thermal hysteresis. We have not identified how peak magnitude varies at very low levels of PMW doping, but away from very low levels of PMW doping, increased PMW content smears the transition, broadening and

[Redacted text and figure]

“For our two compositions, we suggest that increased PMW content smears the transition, broadening and shortening the dielectric peaks.”

The zero-field entropy change, which corresponds to the latent heat of the transition, is reduced with increased PMW doping. This follows from the literature on PST, which shows that better B-site order implies larger latent heat³⁸. This trend should now be clear from what we say in our revised paragraph 3, i.e. that:

“PMW disruption of PST dipolar order suppresses T_C without overly disrupting B-site order, such that the latent heat of the ferroelectric phase transition remains large³⁸”.

The authors simply ignored the requests on characterization of the microstructure from referee 2 (as well as referee 3). The poor quality of TEM (i.e. Fig. 1d) has been questioned according to referee 2 previous comment. Instead, TEM results with much better images are often reported in ferroelectric communities. For comparison, please refer to the EC results in modified PST (Fig. 2 in R. Yin et al., J. Adv. Ceram. 14, 9221088 (2025)) and PMW (Fig. 1 in R. Yin et al., Adv. Funct. Mater. 2502550 (2025)) whereas the quality of TEM is superior to that reported in current work. Now HRTEM is accessible to most labs in the world. TEM with high quality is highly relevant to this work as it inspires confidence in current analysis and quality of current work not to mention that the origin of order/disorder and

microstructure related to increased thermal hysteresis due to including PMW remain unclear.

The TEM that we present is sufficient for our suggestion of B-site dopant segregation. Further TEM work would be interesting, but it is not required to support our main claim of developing materials that can cool a load through room temperature.

The variation of thermal hysteresis with PMW content was explained above.

The authors labour the point that “weak relaxor behaviour (Supplementary Note 12) may be ignored” and “a weak relaxor behaviour notwithstanding” as ““Indirectly measured values of $|Q|$ (solid lines, Fig. 3b) match well with the directly measured values” and ““Indirectly measured values of $|Q|$ (solid lines, Fig. 3b) match well with the directly measured values”. The authors used Q rather than the temperature change. Note that the direct method based on thermocouple measures directly the temperature instead of the Q . The conversion of Q to the temperature requires the heat capacity that is electric field dependent. However, the field-dependent heat capacity has been ignored by authors while its role in analyzing the consistency between indirect and direct methods was not discussed in detail.

As stated earlier, we no longer claim in the main paper or the Supplementary Information that relaxor behaviour is weak. We have thus revised this in the main paper:

“Indirectly measured values of $|Q|$ (solid lines, Fig. 3b) match well with directly measured values, such that weak relaxor behaviour (Supplementary Note 12) may be ignored.”

to:

“Indirectly measured values of $|Q|$ (solid lines, Fig. 3b) match well with directly measured values despite relaxor behaviour (Supplementary Note 5)”.

To say that we “used Q rather than the temperature change” is incorrect because we show temperature change data that is measured with a thermocouple (Fig. 3d). The thermocouple is mentioned in the introduction, in the results section on EC measurements, and in Methods.

To suggest that we have converted heat and temperature data while ignoring the field-dependence of the heat capacity is incorrect because we made no such conversion. The conversions we show are between indirect measurements and direct measurements

of heat (Fig. 3b), and between indirect measurements and direct measurements of temperature change (Fig. 3d). We convert indirect measurements of EC temperature change ΔT (Fig. 2c) to EC entropy change ΔS (Fig. 2f) (which readily yields EC heat $T\Delta S$) via an entropy map (Fig. 2e). By using such an entropy map, the field dependence of the heat capacity is not just not neglected, it can be mapped (as an overlapping set of the current authors have shown in the bottom panels of Fig. 3 in ref. 39).

Moreover, the referee found that the claimed good agreement between indirect and direct method is also conditional. For instance, “A least squares fit between the values of $|\Delta T_j(T_s)|$ that we measured when removing $E = 8.3 \text{ V } \mu\text{m}^{-1}$, and the corresponding values of $|\Delta T(T_s)|$ (solid lines, Fig. S10d, right axis) that we identified for the active volume using the indirect method (Fig.S4c), implies $|\Delta T_j| \sim f |\Delta T|$ with $f = 0.76$. Here, f is the product of a factor f_1 for initial layer thermalization in the active area, and a factor f_2 for subsequent thermalization of the active area with the thermocouple and the affixing drop of black paint.” Following Fig. 3b-3d, the readers might believe that good agreement between indirect and direct methods is achieved without further reading how the fitting technique was imposed. These results are obtained by good fitting with a calibration factor f . The referee may treat that they are not directly consistent with each other. In addition, the measurement frequency for PE loops is not found. The frequency affects PE loops and therefore indirect measurement results. For instance, PE loops measured at high frequency may cover the loss. The PE loops at 230 K are not complete, indicating the presence of leakage. Therefore, the claim that ““Indirectly measured values of $|Q|$ (solid lines, Fig. 3b) match well with the directly measured values” and ““Indirectly measured values of $|Q|$ (solid lines, Fig. 3b) match well with the directly measured values” cannot be used to explicitly rule out the presence of relaxor behavior as the indirect method itself raises concerns to be technically verified. It is suggested to measure the EC response in bulk PST-PMW ceramics via both direct and indirect method (including field-dependent heat capacity) to rule out the inactive volume.

It is incorrect to suggest that our use of a fit implies a hidden inconsistency. In Fig. 3d, the right-axis for indirectly measured adiabatic temperature change is compared with the left-axis for directly measured temperature change in order to account for thermalization with inactive material. Some of the inactive material (inactive ceramic and electrode layers) introduces a known scaling factor of 0.85, while some of the inactive material (thermocouple and affixing drop of black paint) introduces an unknown scaling factor. Our fitting reveals this unknown factor to also be 0.85. Given

that the adiabatic temperature change deduced from the indirect method cannot be realised in practice due to thermalization, scaling for comparison with directly measured data is mandatory. This scaling that we describe above was explained in the main paper.

To say that “the measurement frequency for PE loops is not found” implies that the Referee did not read our short Method section on “Highly adiabatic measurements of electrical polarization”, where we stated that the duration of each unidirectional voltage sweep is <1.3 s (frequency is not specified because, as stated in that Methods section, we set constant currents).

Incomplete $P(E)$ loops at 230 K do not indicate leakage here:

- Any significant leakage would produce Joule heating and preclude a return to the set temperature while the field is on during direct measurements of temperature change. We observe this return at all temperatures, including the much higher temperature of 283 K (inset of Fig. 3d), where any leakage would be stronger.
- To explain the incomplete loops, we have added to the following text in Supplementary Note 7: “Although not relevant for our indirect EC measurements, we note that loop-closure failure below T_C is a consequence of the fact that initial and final states differ (this would be averted in subsequent loops).”.

As explained earlier, we no longer use any match between direct and indirect EC measurements to rule out relaxor behaviour, but as also explained earlier, the indirect method has been correctly applied.

Measuring direct and indirect EC effects in bulk PST-PMW ceramics is not relevant here.

Referee #3 (Remarks to the Author):

Overall, I am satisfied with the response of Guo et al. to my queries and the changes they have implemented as a result. I have read through the revised manuscript and their additional clarifications and explanations are of benefit to the reader for understanding the main results and their potential impact for the electrocalorics field. I think their manuscript is now suitable for publication in its present form.

We thank the Referee for the positive comments and the recommendation to publish in *Nature*. Further changes are described in our response to Referee 2.

Referee #4 (Remarks to the Author):

I co-reviewed this manuscript with one of the reviewers who provided the listed reports.

We thank the Referee for co-reviewing our work and for the recommendation to publish in *Nature*. Further changes are described in our response to Referee 2.

Referee #5 (Remarks to the Author):

I co-reviewed this manuscript with one of the reviewers who provided the listed reports.

We thank the Referee for co-reviewing our work. Further changes are described in our response to Referee 2.